# Spatial variability of $CO_2$ uptake in polygonal tundra – assessing low-frequency disturbances in eddy covariance flux estimates

Norbert Pirk[1,2], Jakob Sievers[3], Jordan Mertes[2,4], Frans-Jan W. Parmentier[5], Mikhail Mastepanov[1,3], and Torben R. Christensen[1,3]

[1]Department of Physical Geography and Ecosystem Science, Lund University, Sölvegatan 12, 22362 Lund, Sweden
[2]Geology Department, The University Centre in Svalbard, UNIS, 9171 Longyearbyen, Norway
[3]Arctic Research Centre, Aarhus University, Denmark
[4]Department of Geological and Mining Engineering and Sciences, Michigan Technological University, 630 Dow Environmental Sciences, 1400 Townsend Drive, Houghton, MI 49931, USA
[5]Department of Arctic and Marine Biology, UiT - The Arctic University of Norway, Postboks 6050 Langnes, 9037 Tromsø, Norway

*Correspondence to:* Norbert Pirk (norbert.pirk@nateko.lu.se)

**Abstract.** The large spatial variability in Arctic tundra complicates the representative assessment of $CO_2$ budgets. Accurate measurements of these heterogeneous landscapes are, however, essential to understand their vulnerability to climate change. We surveyed a polygonal tundra lowland on Svalbard with a UAV, mapping ice-wedge morphology to complement eddy covariance (EC) flux measurements of $CO_2$. The analysis of spectral distributions showed that conventional EC methods do not accurately capture the turbulent $CO_2$ exchange with the spatially heterogeneous surface which typically features small flux magnitudes. Non-local (low-frequency) flux contributions were especially pronounced during snowmelt and introduced a large bias of $-46$ gC m$^{-2}$ to the annual $CO_2$ budget in conventional methods (minus-sign indicates a higher uptake by the ecosystem). Our improved flux calculations with the ogive optimization method indicated that the site was a strong sink for $CO_2$ in 2015 ($-82$ gC m$^{-2}$) and due to differences in light-use efficiency, wetter areas with low-centered polygons sequestered 47% more $CO_2$ than drier areas with flat-centered polygons. While Svalbard has experienced a strong increase in mean annual air temperature of more than 2 K in the last few decades, historical aerial photographs from the site indicated stable ice-wedge morphology over the last seven decades. Apparently, warming has thus far not been sufficient to initiate strong ice-wedge degradation, possibly due to the absence of extreme heat episodes in the maritime climate on Svalbard. However, in Arctic regions where ice-wedge degradation has already initiated the associated drying of landscapes, our results suggest a weakening of the $CO_2$ sink of polygonal tundra.

## 1 Introduction

Arctic tundra is often covered with polygonal ground patterns created by sub-surface ice wedges (*Leffingwell*, 1915; *Mackay*, 1974; *Romanovskii*, 1985; *Minke et al.*, 2007). While ice wedges need centuries or millennia to form through the infiltration and refreezing of meltwater in thermal contraction cracks, they have been reported to degrade rapidly during the last decades with permafrost warming, which significantly alters soil drainage and moisture (*Fortier et al.*, 2007; *Liljedahl et al.*, 2016).

Especially in the high Arctic where permafrost warming occurs fastest (*Romanovsky et al.*, 2010), these hydrological changes can influence the land-atmosphere exchange of greenhouse gases such as carbon dioxide ($CO_2$), which could affect the long-term carbon sink function of these ecosystems (*Schuur et al.*, 2015; *Jorgenson et al.*, 2015).

Carbon dioxide fluxes in polygonal tundra are characterized by large spatial variability, which complicates their accurate assessment over larger areas (*McGuire et al.*, 2012). On a scale of hectares to $km^2$, the micro-meteorological eddy covariance (EC) technique has become widely used to measure the net ecosystem exchange of $CO_2$ (NEE), because it provides a good compromise between directness of measurement, ecosystem disturbance, and technical reliability. The EC technique estimates NEE integrated over a footprint area upwind based on point measurements of the covariance of vertical wind speed and $CO_2$ concentration (*Aubinet et al.*, 2012). Careful calculations have been found to provide defensible estimates of the true $CO_2$ flux, with the main systematic uncertainties stemming from non-steady atmospheric conditions, heterogeneous surfaces and complex terrain (*Baldocchi*, 2003). Large-scale surface heterogeneity has been observed and simulated to induce thermal circulations on the mesoscale that can impede the turbulent flux estimation (*Mahrt et al.*, 1994; *Inagaki et al.*, 2006), while complex terrain may lead to horizontal advection of gases and thereby biased flux estimations (*Finnigan et al.*, 2003; *Aubinet et al.*, 2010). *Finnigan et al.* (2003) showed that the averaging operation and coordinate rotation commonly applied in EC flux calculations can lead to co-spectral distortions and a loss of flux in measurements over tall canopies (forests), even though it is not evident that the same holds for the short-statured vegetation of the tundra where one measures well above the roughness sublayer. Other difficulties relate to the employed measurement instruments, such as when open-path gas analyzers are used, which can introduce a bias due to surface heating of the instrument itself (*Burba et al.*, 2008). Results from different conventionally used software packages for EC calculations have been shown to agree in temperate grasslands and forests (*Fratini and Mauder*, 2014), but the typically low flux magnitudes in high Arctic environments pose considerable challenges for the correct estimation of EC fluxes (*Sievers et al.*, 2015a). Special care must be taken during the long cold season, when small $CO_2$ releases can sum up to a significant portion of the total annual carbon budget (*Fahnestock et al.*, 1999; *Björkman et al.*, 2010; *Lüers et al.*, 2014).

Across the Arctic tundra, previous studies have shown differing $CO_2$ budgets depending on the characteristics of the landscape. EC $CO_2$ measurements from a well studied high Arctic tundra site in Northeast Greenland indicate a considerable annual carbon sink ($-64$ gC m$^{-2}$) in a wet fen (*Soegaard and Nordstroem*, 1999), while the neighboring dry heath constitutes a weaker sink on average ($-21$ gC m$^{-2}$) (*Lund et al.*, 2012). Measurements from Alaskan tussock tundra show that non-growing season releases of $CO_2$ can also exceed the growing season uptake rendering the ecosystem a small net annual source ($+14$ gC m$^{-2}$) (*Oechel et al.*, 2014). A wet tussock grassland in NE Siberia was found to be a moderate annual sink ($-38$ gC m$^{-2}$) (*Corradi et al.*, 2005), while wet polygonal tundra in the Lena River Delta was estimated to be a weaker annual $CO_2$ sink ($-19$ gC m$^{-2}$) (*Kutzbach et al.*, 2007). Some of these studies rely on modeled fluxes to fill large data gaps during wintertime, which increases the uncertainty of the annual sums.

As opposed to other permafrost-underlain regions where soils could become wetter in the future (*Natali et al.*, 2011; *Johansson et al.*, 2013), polygonal tundra is predicted to dry upon permafrost degradation (*Liljedahl et al.*, 2016): the ground above melting ice wedges subsides, which interconnects the polygon troughs and creates an effective drainage network for the wet polygon centers (*Fortier et al.*, 2007). This simultaneous wetting of polygon troughs and drying of polygon centers is a signa-

ture that can be detected in time series of historical aerial photographs to provide large-scale evidence of the process (*Necsoiu et al.*, 2013). *Liljedahl et al.* (2016) described such ice-wedge degradation as a widespread Arctic phenomenon, which changed surface drainage patterns in less than one decade. To study the progressive change in plant communities and biogeochemistry following such a disturbance space-for-time substitutions have proven to be a powerful tool (*Rastetter*, 1996). The associated hydrological changes have thus been linked to significant changes in carbon fluxes in Alaskan polygonal tundra during the growing season (*Lara et al.*, 2015; *Vaughn et al.*, 2016), but their year-round effect on an ecosystem's $CO_2$ budget is yet to be quantified.

Such an assessment could be improved by high-resolution topographical surveys using unmanned aerial vehicles (UAVs). Apart from the visual picture UAVs provide, the series of photographs from multiple angles allows the reconstruction of the 3D geometry of the surface (*Ullman*, 1979; *Westoby et al.*, 2012), which can give valuable insights into the drainage patterns. In the present study, we explore the potential of this technique in combination with EC $CO_2$ flux measurements in polygonal tundra on Svalbard to characterize the spatial heterogeneity of the ecosystem. We aim to understand how the spatial heterogeneity and larger-scale disturbances affect EC flux estimates by investigating the spectral composition of the EC signal. We further relate the spatial differences in NEE to the observed historical, and the predicted future evolution of ice-wedge polygons.

## 2 Materials and Methods

### 2.1 Site description

The field site is located at the bottom of a large, permafrost-underlain, glacial valley called Adventdalen on Spitsbergen, Svalbard, approximately 6 km from a fjord (78°11′N, 15°55′E). The surrounding mountains feature plateaus of around 450 m a.s.l., as well as peaks and ridges of up to 1000 m a.s.l, which are still partly glaciated (*De Haas et al.*, 2015). Wind directions are generally oriented along the valley, with dominating easterlies in wintertime (coming from inland Spitsbergen), and an approximately even distribution of easterlies and westerlies in summertime (westerlies coming from the fjord). Long-term statistics indicate that wind speeds in Adventdalen are below 5 m s$^{-1}$ for about 70% of the year (and below 10 m s$^{-1}$ for about 97%), with a most frequent wind speed of about 3 m s$^{-1}$. The mean annual air temperature at the closest weather station (Svalbard airport, approximately 10 km away) was $-6.7$°C between 1961 and 1990 (*Førland et al.*, 2012), which has increased to $-3.75$°C in the period between 2000 and 2011 (*Christiansen et al.*, 2013). The total annual precipitation is about 190 mm, of which about half falls as snow (*Førland et al.*, 2012). The measurement site is located on a river terrace on the flat part of a large alluvial fan, where the ground is patterned by ice-wedge polygons (*Christiansen*, 2005; *Harris et al.*, 2009). These coarse alluvial deposits are covered with a few ten centimeters of organic material and fine-grained eolian deposits (loess), which typically stem from wind erosion in the braided riverbed when it dries out in autumn (*Bryant*, 1982; *Oliva et al.*, 2014). The site's soil organic carbon content in the uppermost 100 cm soil is about 30 kgC m$^{-2}$ (*personal communication with Peter Kuhry*). The vegetation at the Adventdalen site features *Salix polaris* in drier areas, *Eriophorum scheuchzeri* and *Carex subspathacea* in wetter locations. The moss cover is sparse in drier polygons where shrubs dominate the vegetation community, while the wetter

areas at local depressions feature an almost continuous moss cover. Within individual polygons the moss coverage typically increases from the drier rim to the wetter center.

## 2.2 Measurement setup

The EC setup consisted of a top-mounted ultra-sonic anemometer (USA-1, Metek GmbH, Germany) and an infra-red gas
analyzer (Li-7200, Li-Cor Inc., USA), both of which were sampling and recording data at a rate of 10 Hz. The measurement height was 2.8 m above ground level. From there the gas was pumped to the Li-7200 at a flow rate of 15 L min$^{-1}$ via a 1 m long, insulated intake tube supplied by the manufacturer.

Ancillary meteorological measurements (e.g. solar radiation, snow and soil temperatures) were collected on and around the same tower, sampled every 10 s and averaged to 30 min values. Due to the relatively remote location without line power, the
system was supplied by lead-acid batteries, which were charged by a wind generator (350 W peak output) and solar panels (275 W peak output), as well as a fuel cell in summertime (90 W).

Complementary to the EC setup, we measured NEE in the EC footprint with a set of five transparent, automatically operated, flux chambers using the closed chamber technique. These chambers were connected to a gas analyzer (SBA-4, PP Systems, UK), which measured $CO_2$ concentrations at a rate of 0.625 Hz. Flux estimates were derived from exponential least-squares
regression of the 5 min closure time of the concentration time series. The details of this measurement system and flux estimation procedure are provided in *Pirk et al.* (2016b) and references therein.

To assess differences in the active layer depth, thaw depths were probed at the centers of 30 polygons in the EC footprint at the end of August 2016.

## 2.3 Data processing

EC flux estimates were derived using the recently proposed ogive optimization method (version 1.0.5, toolbox publicly available through the Mathworks file exchange) (*Sievers et al.*, 2015b). In this context, ogives are cumulative co-spectra of vertical wind speed (w) and $CO_2$ concentration (denoted Og(w,$CO_2$)), i.e. a spectral decomposition of the EC flux estimate. The method optimizes a spectral distribution model (*Desjardins et al.*, 1989; *Lee et al.*, 2006; *Foken et al.*, 2006) to a density map of 14 000 ogives obtained by varying the dataset length and de-trending interval. The key to this method is the assumption of a dynamic
spectral gap between often overlapping spectral flux contributions (*Sievers et al.*, 2015b). This approach effectively separates the turbulent flux from contributions of larger-scale motions (mesoscale atmospheric movements), which can give non-local flux contributions at low frequencies (*Aubinet et al.*, 2012; *Sievers et al.*, 2015b).

To further investigate the effect of low-frequency contributions we compared ogive optimization to the widely used EddyPro software package (Li-Cor Inc., version 6.1.0), following the conventional assumption about the presence of a fixed spectral
gap corresponding to the 30 min flux averaging interval. We used simple linear de-trending, and applied spectral corrections according to *Moncrieff et al.* (1997, 2004) (EddyPro default).

Both EddyPro and ogive optimization perform basic quality control and pre-processing of the 10 Hz raw data following *Vickers and Mahrt* (1997) (e.g. gap detection, spike removal, signal alignment, anemometer tilt correction). Unacceptable raw

data were not processed further. To ensure sufficient turbulent mixing near the surface, we also filtered out data points with a friction velocity smaller than $0.1 \, \mathrm{m \, s^{-1}}$ for both methods. Ogive optimization furthermore only accepts periods with a negative momentum flux (i.e. directed toward the ground surface) in the mid-frequency range (tested at around 0.032 Hz), which is the energy-containing range. Following *Foken and Wichura* (1996), EddyPro fluxes were additionally filtered for non-steady wind conditions (discarding fluxes with quality flag 2). Calculated ogive optimization fluxes, on the other hand, were only discarded if the modeled ogive spectral distribution could not describe the data sufficiently well. These filters, in addition to down-time caused by technical problems, led to an overall data coverage with valid fluxes of 45% in 2015 for the ogive optimization and 35% with EddyPro. A large number of these flux calculations have been visually inspected to ensure that the methods performed as expected. In this analysis, we noticed that the automatically determined time lags between w and $CO_2$ concentration varied unrealistically, which introduced noise to the fluxes—especially at low magnitudes. We therefore used a constant value of 0.3 s (i.e. the typically expected time lag given our setup) for the flux calculation with both methods.

We mainly focused on data collected between September 2014 and December 2015, when data quality and coverage was highest. Carbon dioxide concentrations collected in 2013 were only recorded as molar densities and without the cell pressure necessary for a sample-by-sample conversion to mixing ratios according to the Webb-Pearman-Leuning correction proposed by *Sahlée et al.* (2008), which is currently the only option implemented in the ogive optimization software. Hence, we only report 2013's fluxes from EddyPro as supplementary support for our findings.

Subsequently, the calculated NEE was used to determine the ecosystem's light response characteristics during the snow-free period (beginning of June until end of September). One way to parameterize the relationship between NEE and incoming photosynthetically active radiation (PAR) is the Misterlich function,

$$\mathrm{NEE} = -\left(F_{\mathrm{csat}} + R_{\mathrm{d}}\right) \left(1 - \exp\left(-\frac{\alpha \, \mathrm{PAR}}{F_{\mathrm{csat}} + R_{\mathrm{d}}}\right)\right) + R_{\mathrm{d}}, \tag{1}$$

where the three parameters $F_{\mathrm{csat}}$, $R_{\mathrm{d}}$, and $\alpha$ correspond to the flux at light saturation, dark respiration, and light-use efficiency, respectively (e.g., *Falge et al.*, 2001). Such light response curves can yield further insights to the underlying drivers of NEE. These parameters were derived from least-squares regressions of measured NEE and PAR (derived from short wave incoming radiation) in a rolling time window of 10 days, which was successively increased by 1 day if less than 100 valid NEE-PAR measurements were available, following *Lund et al.* (2012).

For cumulative flux calculations and annual sums we employed the gap-filling algorithm proposed by *Reichstein et al.* (2005), which operates on the basis of mean diurnal variations of temperature, incoming short wave radiation, and vapor pressure deficit as drivers for NEE. As some of the gaps in our NEE measurements were caused by power outages (when the entire measurement system shut down), the ancillary data for gap-filling were taken from the New Adventdalen Weather Station (run by the University Centre in Svalbard) with a distance of about 2.5 km from our site. This procedure yielded the best gap-filling quality (class A) in 96% of the flux estimates that had to be gap-filled in 2015. Still, gaps in NEE measurements can be assumed to dominate the total random error of the annual sums (*Aurela et al.*, 2002). To quantify this uncertainty, we tested the sensitivity of the annual sums to artificially added gaps in the NEE time series. Since the uncertainty introduced by a gap depends on its length and time of year, we repeated the gap-filling on 300 different time series obtained by adding single

gaps with a length between 1 and 23 days (i.e. 2 days longer than our longest gap) which were equally distributed over the year (starting every 15 days). The resulting distribution of annual sums was used to assess the result's random error.

The EC footprint estimation was performed according to *Kljun et al.* (2015), using a fixed zero-plane displacement of 10 cm, and a roughness length of 1 cm. Wind and turbulence parameters were derived from 30 min intervals, while the additionally needed boundary layer height was taken from the closest point of the Era-Interim meteorological reanalysis (*Dee et al.*, 2011).

## 2.4 Topographical survey

We conducted a topographical survey of the Adventdalen ice-wedge site employing photogrammetry of aerial photographs to produce a visual map and digital elevation model. To this end, 135 photographs were taken with a camera (GoPro Hero3+ Black Edition) from a UAV (DJI Flamewheel 550) at a height of 60 m a.g.l. in June 2015. This survey covered an area of about 0.1 km$^2$ at a ground resolution of about 3.2 cm pixel$^{-1}$. 22 ground control points were collected with a differential GPS (Leica GPS1200 SmartRover) to ortho-rectify the images and estimate the uncertainty of the resulting elevation model (see Fig. S5 in the supporting information for details). GPS data were post-processed and differentially corrected using data from the local Longyearbyen GNSS satellite basestation (LYRS), which is freely available from the Norwegian Mapping Authority. The photogrammetric processing was performed using Agisoft PhotoScan (Agisoft LLC, St. Petersburg, Russia), which implements the structure-from-motion technique to reconstruct the 3D geometry of the ground surface from the sequence of photographs taken from multiple viewpoints (*Ullman*, 1979; *Snavely et al.*, 2008; *Westoby et al.*, 2012; *Lucieer et al.*, 2014).

To assess the evolution of the morphology of the ice-wedge site, we compared our map from 2015 to historical aerial photographs taken 1948, 1961 and 1990, which were geo-referenced with 2015's map as a reference. Reliably quantifying the changes between these images was complicated by different shadows and overall soil moisture when the images were taken, so we only performed a qualitative change detection by visual comparison.

## 3 Results

Figure 1 shows the results of the topographical survey and the EC footprint for June 2015. As the wind direction typically aligns with the valley's direction, there are two clearly distinct footprint areas in the NW and ESE. The high resolution of this elevation map resolves small elevation differences of only a few decimeters, which can be seen to affect surface inundation and soil wetness. Both footprints have overall surface slopes toward the edge of the river terrace in the north (approximately 0.75% slope), but their drainage patterns appear to be separated creating a wetter sub-catchment in the NW than in the ESE. Low-centered polygons dominate the site, but the NW features more distinctly wet polygon centers. The thaw depth at the centers of the polygons around the EC tower was 66 cm $\pm$9 cm (mean $\pm$standard deviation, sample size N=30) by the end of August. Based on the polygons in the 50% EC footprint, the drier ESE fetch area featured a thaw depth of 69 cm $\pm$8 cm (N=4) while the wetter NW featured 79 cm $\pm$4 cm (N=4), which is not a statistically significant difference (p=0.10).

The shown surface heterogeneity is likely to lead to spatial variations of NEE in the EC footprint. To assess this effect and mesoscale disturbances, we investigated the spectral composition of the EC signal by looking at the ogives of vertical

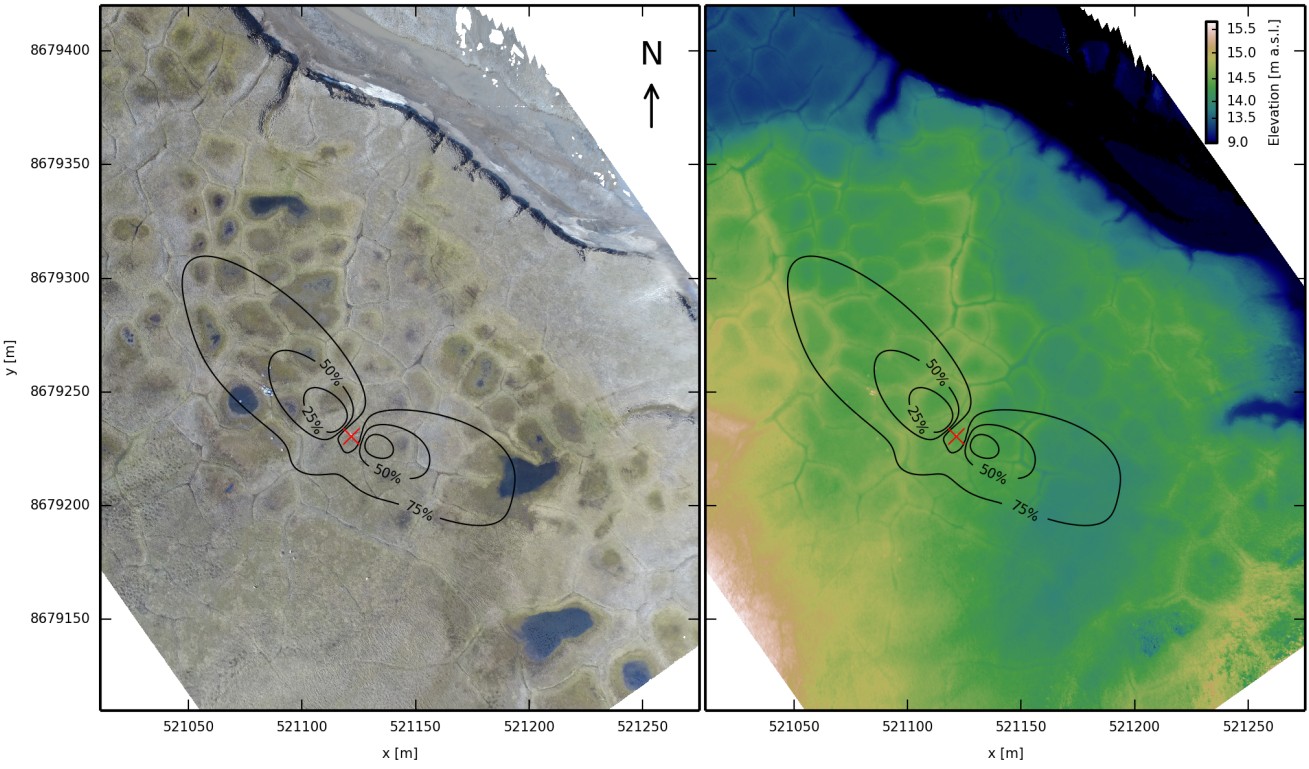

**Figure 1.** Map of the site in Adventdalen (coordinates in UTM zone 33X). The red cross marks the EC tower, around which the contour lines indicate the area's relative contribution to the EC signal (footprints) averaged over June 2015. Six automatic flux chambers are located in the NW footprint (bright dots). Left: Ortho-rectified aerial photograph from end of June 2015. Right: Corresponding surface elevation with an estimated vertical uncertainty of 0.2 m.

wind speed and $CO_2$ concentration. Particularly around the time of snowmelt, we often found a mismatch between lower and higher frequencies indicating different local and non-local flux contributions. Figure 2 shows an example from this period comparing the ogives of conventional flux calculations produced by EddyPro and ogive optimization flux estimation based on the ogive density map. While all frequencies contribute fully to the conventional flux estimation, frequencies can obtain less weight in the ogive optimization method if they cannot be described by the ogive spectral distribution model. In the given example, this conceptual difference of the methods means that ogive optimization indicated a $CO_2$ release, while EddyPro indicated uptake. Spectral corrections had a comparably small effect. The ogive optimization model indicates that all relevant flux contributions are carried by turbulence with a scale shorter than about 25 s in this example (which does, however, not mean that 25 s are enough to determine a 30 min flux). During this period in May and June the surface was a mix of patches of snow-free soil, remaining snow, and meltwater ponds (when a net $CO_2$ uptake can be considered unlikely, see Fig. 2b). The low-frequency contributions also depend on larger-scale atmospheric movements, while the local turbulent flux is represented

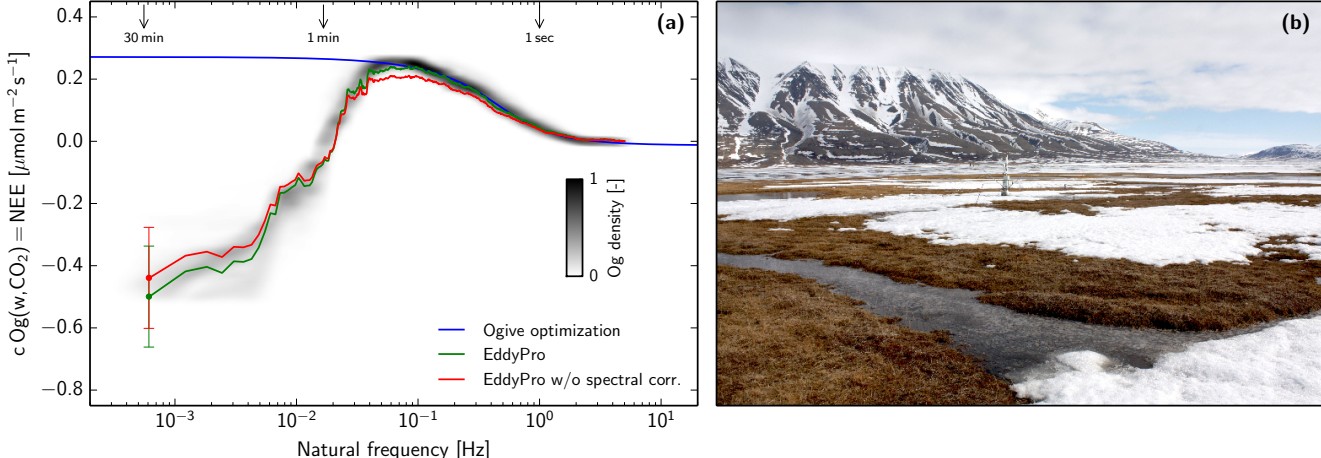

**Figure 2.** Example of the $CO_2$ flux estimation during snowmelt. a: Ogive on 31 May 2015, 15:00 LT, showing a mismatch between low and high frequencies. While ogive optimization estimates a net $CO_2$ release, EddyPro (with and without spectral corrections after *Moncrieff et al.* (1997, 2004)) indicates an uptake. Average horizontal wind speed was 5.2 m s$^{-1}$ from NW (313°), air temperature 4.5°C, quality flag 0. Arrows on the top indicate the corresponding time scales. b: Photo of the environment around the flux tower on 27 May 2015, 12:00 LT during snowmelt.

better in the mid-to-high frequency range. Such frequency mismatches (high-frequency $CO_2$ release, but low-frequency uptake) were frequently observed in our data and their effect was relatively largest for the small non-growing season fluxes (see Fig. S1 in the supporting information for additional ogive examples and Fig. S3a for a flux comparison). The shifts between release and uptake of $CO_2$ typically occurred at frequencies below $10^{-1}$ Hz, corresponding to recorded eddies with a diameter of typically more than 30 m at the given wind speed. During the growing season, fluxes from both methods typically agreed. Photos of the site in different seasons are given in Fig. S6 in the supporting information.

Since the frequency mismatches cannot be resolved in conventional calculations, we focus on the NEE fluxes calculated by the ogive optimization method for the ecosystem characterization. Figure 3 shows the gap-filled NEE fluxes of 2015 as fingerprint plots, as well as cumulative sums, which were also calculated after separately gap-filling the measurements of the two distinct footprints (NW and ESE). These results indicate that the growing season in 2015 started on 14 June and ended 28 August (defined as first until last day of net $CO_2$ uptake). Results from EddyPro indicated the same date for the end of the growing season, while its start was suggested already one month before snowmelt. The fingerprint plot (Fig. 3a) shows that there can even be $CO_2$ uptake at midnight during the polar day in the summer. While the drier ESE yielded an annual carbon balance of $-62$ gC m$^{-2}$, the wetter NW yielded $-91$ gC m$^{-2}$. The annual balance of the combined footprint (using all fluxes without separating periods of different wind directions) was $-82$ gC m$^{-2}$ in 2015. The corresponding value based on EddyPro flux calculations was $-128$ gC m$^{-2}$, which we consider biased by the above-mentioned low-frequency contributions. The relatively narrow probability distributions of the annual sums (based on gap-filling uncertainties) demonstrate the significance

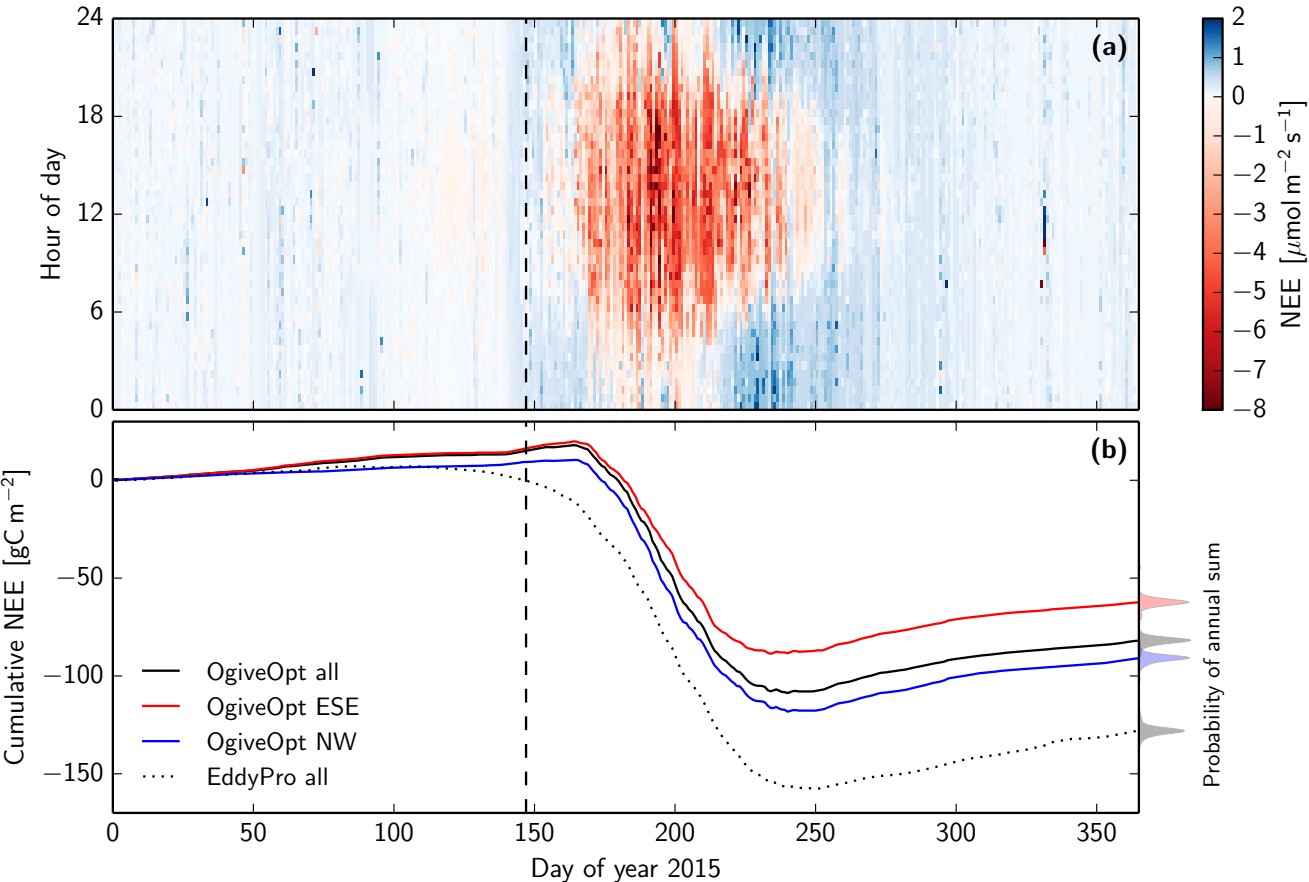

**Figure 3.** Gap-filled NEE fluxes for 2015. a: Fingerprint plot of ogive optimization results. b: Corresponding cumulative sums based on all valid measurements (black), and separately gap-filled for the two footprints (colored). The probability distributions shown on the right indicate the estimated uncertainty of the annual sum due to data gaps and gap-filling. The dashed line marks the time during snowmelt when daily average albedo dropped below 0.3 (27 May 2015).

of the differences between the NW and ESE footprints, as well as between the ogive optimization and EddyPro methods. Relatively large annual sinks are supported by our automatic closed chamber measurements in the NW footprint which show good agreement and correlation ($0.75 < r < 0.88$, $p < 0.0001$) with the EC fluxes (see Fig. S3b in the supporting information). In 2013, EddyPro calculations yielded a smaller total annual $CO_2$ balance of $-79$ gC m$^{-2}$ (see Fig. S2d in the supporting information), whereas ogive optimization fluxes could not be calculated from 2013's raw $CO_2$ measurements (cf. section 2.3).

Much of the spatial differences in the annual $CO_2$ budget stem from the growing season, when NEE is strongly affected by PAR. Figure 4 shows examples of the derived light response curves, as well as the evolution of the associated dark respiration and light-use efficiency throughout the growing season. Both dark respiration and light-use efficiency were typically higher in the wetter footprint (NW) than in the drier (ESE), consistent with the larger annual uptake in the NW than the ESE. At

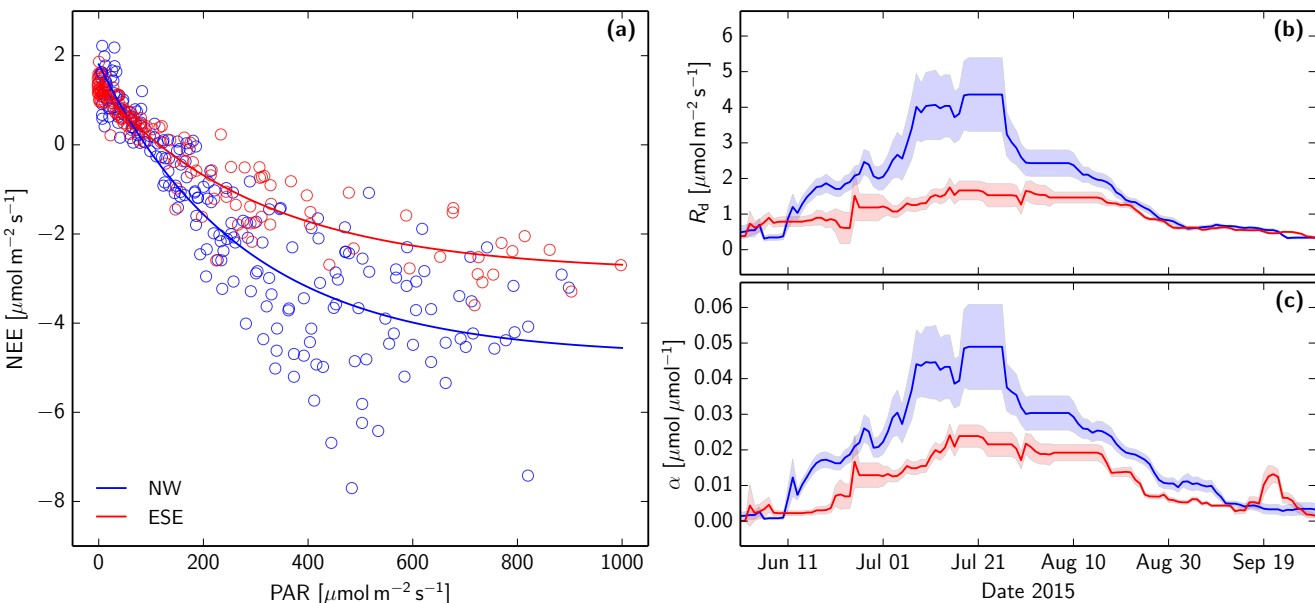

**Figure 4.** Growing season NEE light response curves based on the ogive optimization method. a: Examples from the time window around 18 August 2015 from the two distinct footprints. b: Time series of dark respiration parameters. c: Corresponding graph for light-use efficiency. Shaded bands indicate the statistical standard error of the parameters.

the beginning and end of the growing season, the determination of the flux at light saturation ($F_{\mathrm{csat}}$) was associated with relatively large uncertainties, because NEE and PAR varied relatively little. During the peak growing season $F_{\mathrm{csat}}$ was about

6.6 $\mu$mol m$^{-2}$ s$^{-1}$ for both footprints. The sum of $F_{\mathrm{csat}}$ and $R_{\mathrm{d}}$ can be used to estimate the gross primary productivity at light saturation, which was found to be $-11.0$ $\mu$mol m$^{-2}$ s$^{-1}$ in the NW and $-8.3$ $\mu$mol m$^{-2}$ s$^{-1}$ in the ESE. During snow covered conditions outside the growing season, our measurements indicated an overall decreasing trend of the small $CO_2$ releases throughout winter, which was modulated by increases during strong winds (see Fig. S4 in the supporting information). Soil temperature, on the other hand, had no strong effect on the wintertime $CO_2$ release despite a large variation of more than 25 K

(cf. Fig. S4c).

Figure 5 shows a time series of ortho-rectified aerial photographs covering the same area as Fig. 1. Despite the differences in image quality, shadows and overall soil moisture on the days these images were taken, the time series still gives an impression about the development of the polygon morphology. All images show the same, low-centered polygons, whose centers were about equally inundated (except in 1990 when the area was drier in general). There was no clear lateral expansion or degradation

of the polygon troughs. Neither the ponds nor the troughs became more interconnected or wet in general, so there are no clear signs of differential ground subsidence at this site. Also other areas with ice-wedge polygons in Adventdalen indicated the same, stable morphology during the last seven decades (see Fig. S7 in the supporting information). Between 1990 and 2015, some erosion on the edge of the river terrace occurred. The exact speed of this edge erosion is hard to quantify due to the

shadows in this area, but it did not exceed 3 m over these 15 years. The photograph from 1990 was taken in the near infra-red range and is shown in false colors. Vegetation, bare soil, and open water reflect near infra-red light differently, so this image clearly depicts these surfaces. The strong red tones in the NW footprint correspond to the relatively high vegetation density in this area, which was also seen in Fig. 1. The river in the NE appears in a blue tone, while darker spots in the SW corner of the image correspond to an area with bare soil brought to the surface by cryoturbation.

## 4   Discussion

Our measurements demonstrate the high sensitivity of carbon cycling to small topographic differences in permafrost-underlain Arctic tundra. Ice-rich permafrost is particularly vulnerable to warming, with large increases in permafrost degradation documented in the last two decades (*Jorgenson et al.*, 2006; *Osterkamp et al.*, 2009; *Grosse et al.*, 2011). *Liljedahl et al.* (2016) observed pan-Arctic permafrost degradation in polygonal tundra which dramatically changed the local drainage patterns and water balance on sub-decadal timescales. Ice-wedge melting and the associated differential ground subsidence is expected to interconnect formerly separated trough networks and thereby increase the drainage of polygon centers (*Necsoiu et al.*, 2013; *Jorgenson et al.*, 2015; *Liljedahl et al.*, 2016). At a later stage this process transforms low-centered polygons into high-centered polygons and leads to the overall drying of the entire landscape. In such cases, the space-for-time substitution of our two distinct footprint areas in Adventdalen would suggest a corresponding lessening of $CO_2$ sinks in degrading polygonal tundra.

However, our comparison of aerial photographs taken between 1948 and 2015 shows that there is no dramatic ice-wedge degradation at our site on Svalbard. Nearby areas with polygonal tundra in Adventdalen have also been stable during the last seven decades, despite the measured increase of $2.95°C$ in mean annual air temperatures between the periods 1961–1990 and 2000–2011 (*Førland et al.*, 2012; *Christiansen et al.*, 2013; *Nordli et al.*, 2014). The maritime climate on Svalbard prevents episodes with extremely high summer temperature, which have been hypothesized to trigger ice-wedge degradation (*Jorgenson et al.*, 2006; *Liljedahl et al.*, 2016). It appears that the gradual climate warming alone has so far not been sufficient to initiate strong ice-wedge degradation in Adventdalen. Another reason for the apparent stability of the ice wedges at our site could be the relatively small surface slope of typically less than 1%, which hinders the development of an effective drainage system of degraded troughs. Generally speaking, ice-wedge stabilization can also be caused by negative feedbacks such as increased plant growth in degraded troughs which cools the soil above the ice wedge (*Jorgenson et al.*, 2015). Despite these mechanisms and observations, the strong temperature increase on Svalbard may eventually lead to the gradual degradation of ice wedges in Adventdalen, which we argue will lessen the $CO_2$ sink of this ecosystem.

The overall annual $CO_2$ balance of $-82$ gC m$^{-2}$ seems surprisingly large, given the high northern location of the site with its typically shallow organic horizon in the soil (5–10 cm). The good agreement of EC and automatic closed chamber measurements, however, confirms the relatively high uptake fluxes in the snow-free period. Also the comparison of the light response curve parameters to other Arctic sites indicates high, but realistic, growing season productivity. *Mbufong et al.* (2014) derived these parameters from 12 Arctic tundra sites during the peak growing season and report $R_d$ between 0.6 and 3.9 $\mu$mol m$^{-2}$ s$^{-1}$, and $\alpha$ between 0.011 and 0.057 $\mu$mol $\mu$mol$^{-1}$. Peak season parameters for the drier footprint (ESE,

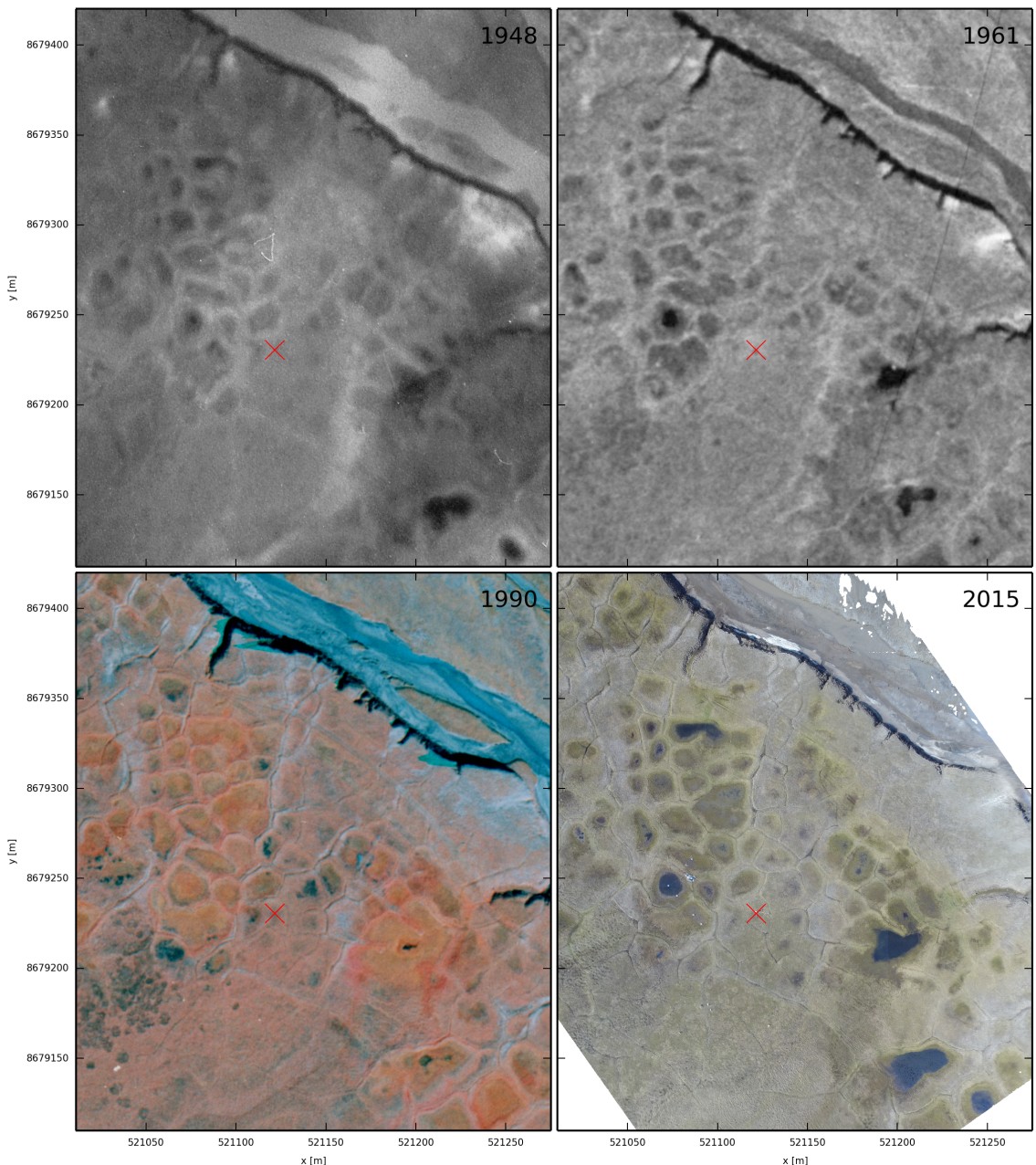

**Figure 5.** Time series of images of the Adventdalen site showing little signs of differential ground subsidence, which would indicate ice-wedge degradation. The image from 2015 is the same as shown in Fig. 1, while historical photographs were provided by the Norwegian Polar Institute (reference numbers S48-5181, S61-3301 and S90-5273). The images from 1948 and 1961 were taken on panchromatic films, and the image from 1990 is a near infra-red (false color) photography. The red cross marks the EC tower.

$R_\mathrm{d} \sim 1.7$ $\mu$mol m$^{-2}$ s$^{-1}$, $\alpha \sim 0.024$ $\mu$mol $\mu$mol$^{-1}$) at the Adventdalen site lie well in the center of this range, while the wetter
footprint (NW, $R_\mathrm{d} \sim 4.3$ $\mu$mol m$^{-2}$ s$^{-1}$, $\alpha \sim 0.049$ $\mu$mol $\mu$mol$^{-1}$) lies on the upper end of this reported range. These values,
in combination with the relatively large amount of incoming shortwave radiation at 78°N during summertime, explain the
large carbon sink in Adventdalen. The summertime $CO_2$ sink of Adventdalen is comparable to this season's $CO_2$ balance at a
coastal wet sedge tundra ecosystem at Barrow, Alaska ($-105$ to $-162$ gC m$^{-2}$) (*Harazono et al.*, 2003). Especially the wetter
areas lose some carbon through methane emissions, but automatic chamber measurements at Adventdalen indicate that these
losses are not expected to exceed 6 gC m$^{-2}$ per year (*Pirk et al.*, 2016a). We consider the export of dissolved organic carbon
negligible, because the small surface slope and limited conductivity prevents a pronounced lateral water runoff.

The grazing pressure from Svalbard reindeer could represent another type of carbon loss, which has not been quantified in the
present study. Moreover, *Wegener and Odasz-Albrigtsen* (1998) observed that plants in Adventdalen balance the consumption
by reindeer with increased plant productivity. While such interactions influence the carbon budget of the ecosystem, they do
not affect the discrepancy we found between the two EC flux calculation methods.

The drivers of cold season emissions of $CO_2$ from high Arctic tundra are still understudied, because technical challenges
and low flux magnitudes often complicate continuous in-situ flux measurements. Our wintertime flux measurements from
Adventdalen were found to decrease slightly throughout winter. Episodic flux increases correlated with wind speed, suggesting
a convective mixing of the snowpack gas reservoir as observed in other studies from lower latitudes (*Takagi et al.*, 2005; *Seok
et al.*, 2009; *Smagin and Shnyrev*, 2015). The missing relation between soil temperature and measured wintertime $CO_2$ release
could suggest a decoupling of $CO_2$ production and release caused by the physical blockage of gas diffusion in the soil (*Elberling
and Brandt*, 2003) or through potential ice layers in the snowpack (*Pirk et al.*, 2016a). While the majority of wintertime fluxes
were positive, some small uptake fluxes have also been observed during the dark, snow covered period. Unlike reports from
other sites (*Lüers et al.*, 2014), these fluxes have no significant impact on the annual $CO_2$ budget of our site. As photosynthesis
by plants or snow algae can be excluded during the dark polar night, one might speculate that the apparent uptakes are caused by
abiotic mechanisms, such as the convective mixing of $CO_2$-depleted gas stored in the snowpack or thermo-physical processes
related to $CO_2$ solubility in unfrozen pore water. Yet we found no relationship between uptake situations and changes in snow,
air or soil temperatures, or ambient atmospheric $CO_2$ concentrations—which could support potential abiotic mechanisms of
$CO_2$ uptake. The magnitude of these fluxes was also so low compared to biotic flux contributions that they cannot markedly
change the overall annual $CO_2$ balance of the ecosystem and remain to be regarded as noise in this study.

The conceptual definition of turbulent fluxes differs fundamentally between the ogive optimization and conventional methods
as implemented in EddyPro. While the ogive optimization assumes a unidirectional flux, which is sometimes better captured
in the mid and high-frequency range, EddyPro includes all frequencies regardless of their direction. The ogive optimization
method appeared better suited for the Adventdalen site than conventional processing schemes. Specifically around the snowmelt
period, the ogive optimization estimates appear to capture the local flux signal more realistically than conventional calculations,
which indicated an onset of the growing season already before snowmelt. This contradiction was caused by many bi-directional
fluxes, i.e. situations with a consistent $CO_2$ release reflected in high frequencies and $CO_2$ uptake reflected in the low-frequency
range of the spectrum (cf. Fig. 2 and Fig. S1). The shift between these contributions occurred at frequencies corresponding

to eddies with a diameter of more than 30 m, i.e. exceeding the typical dimension of surface heterogeneity in the footprint

area of our site. However, we cannot fully exclude that the frequency mismatches are caused by flux heterogeneity within the local footprint. It could be possible that the drier areas near the EC tower (reflected better in the high frequencies) are net $CO_2$ sources, while wetter areas at larger distances from the tower (reflected in the low frequencies) are net $CO_2$ sinks. A heterogeneous vegetation composition might cause such flux heterogeneity during snowmelt, because unlike shrubs and sedges, mosses have photosynthetically active tissue that may overwinter so that they can start photosynthesizing at low rates

already during snowmelt (*Oechel*, 1976; *Tieszen et al.*, 1980). While some degree flux heterogeneity is certainly present at any time of year, its effect might be too small to explain the large frequency mismatches observed particularly during snowmelt. Nevertheless, our observations might incentivize future studies to investigate the frequency dependency of EC footprints. Snowmelt also entails a change in the typical surface roughness length, which is slightly smaller for snow than open tundra vegetation. However, such shifts would be no problem for the functionality of the used flux calculation schemes and cannot

readily explain bi-directional fluxes even if the surface roughness is spatially heterogeneous. One might speculate that the systematic occurrences of bi-directional fluxes are due to an atmospheric layering where low and high-frequency eddies circle through air masses with different atmospheric stability and $CO_2$ concentrations: While the (smaller) high-frequency eddies only reflect one air mass, the (larger) low-frequency eddies reflect two air masses with different $CO_2$ concentrations. In the specific environment in Adventdalen, such a layering might be induced by the intrusion of $CO_2$-depleted air at the surface

originating from the surrounding mountains by way of katabatic winds, or sea breeze circulations from the nearby fjord (*Esau and Repina*, 2012). However, similar low-frequency shifts were also observed in environments with neither pronounced surface heterogeneity nor nearby water bodies, snow or mountains (*Sievers et al.*, 2015b), as well as in temperate regions (*Finnigan et al.*, 2003). Across the different sites we see a tendency for low-frequency shifts to occur predominantly during conditions with small flux magnitudes, when the normally dominating mid and high-frequency contributions can be much smaller than

(non-local) low-frequency contributions. So perhaps the hypothetical layering of the atmosphere is caused by the repeatedly changing $CO_2$ flux at the surface in response to diurnal factors such as changes in incoming solar radiation, which give rise to $CO_2$ concentration waves propagating vertically into the atmosphere. While such hypotheses remain to be investigated in future studies, we show that these (non-local) low-frequency contributions lead to a difference in the annual $CO_2$ budget of $-46$ gC m$^{-2}$ over the course of one year (cf. Fig. 3b). The ogive optimization method is more applicable to these highly

heterogeneous, Arctic environments dominated by small fluxes because it can separate local and non-local flux contributions.

## 5   Conclusions

The Adventdalen ice-wedge site was a surprisingly strong $CO_2$ sink in 2015 ($-82$ gC m$^{-2}$). Differences in vegetation density and composition lead to a significantly higher light-use efficiency in areas with low-centered ice-wedge polygons compared to flat-centered polygons. While dark respiration in the wetter area was also higher than in the drier area, these releases did

not compensate for the higher light-use efficiency in the annual $CO_2$ balance. In 2015, the drier area sequestered 32% less $CO_2$ than the wetter area ($-62$ compared to $-91$ gC m$^{-2}$). These results suggest a high sensitivity of $CO_2$ dynamics to small

topographic differences in Arctic tundra ecosystems. With climate warming, ice wedges are predicted to melt and dry out the landscape. Despite strong increases in mean annual air temperatures of more than 2°C on Svalbard in the last few decades, we see no evidence of ice-wedge degradation compared to historical aerial images. However, further warming may eventually initiate ice-wedge degradation, and our spatial analysis implies a corresponding reduction of the $CO_2$ sink upon drying. In Arctic polygonal tundra where drying is occurring already, our results therefore suggest a similar weakening of the $CO_2$ sink function.

## 6   Data availability

Maps, measurement data and processing scripts are available from the authors upon request (norbert.pirk@nateko.lu.se).

*Competing interests.* The authors declare that they have no conflict of interest.

*Acknowledgements.* The research leading to these results has received funding from the European Community's Seventh Framework Program (FP7) under Grants 238366, 262693 and 282700, and the Nordic Centers of Excellence DEFROST and eSTICC (eScience Tool for Investigating Climate Change in northern high latitudes) funded by Nordforsk (grant 57001). We thank Sarah Strand and Andreas Alexander (UNIS) for their work at the Adventdalen site, and Sebastian Westermann (University of Oslo) for providing computational resources needed for the data processing.

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
