# Peer review of "Spatial variability of CO2 uptake in polygonal tundra – large overestimations by the conventional eddy covariance method"

_Biogeosciences, 2016_

## Referee Comment (RC1) · W. Eugster (Referee) · 7 Feb 2017

W. Eugster (Referee)

werner.eugster@usys.ethz.ch

The authors measured net CO$_2$ fluxes over polygonal tundra in the high arctic on Svalbard. They present an interesting and well written manuscript, but seem to interpret some aspects differently than this reviewer. I think it is important to exchange ideas and opinions between authors and reviewers and that it is positive to have new ideas presented even if not everyone agrees. However, my main points given below is that the language should reflect this in a somewhat clearer way so that the uninitiated reader does not misinterpret the universality of some statements.

Having said this, I must admit that I learned a lot reading this manuscript and fully support its publication after careful revisions.

[Figure]

Besides an important methodological aspect of how to compute defensible annual flux sums, a key statement of the paper is that all the detailed image analyses starting with pictures taken in 1948 cannot confirm a rapid degradation of this polygonal tundra, but rather support the view that this landscape has been quite stable over the last seven decades.

**Main critique**

1. Your introduction completely misguided me in the wrong direction as your paper starts with the phrase "Carbon-rich Arctic tundra soils are often covered with polygonal ground patterns created by sub-surface ice wedges." – (1) Your paper is **not** addressing carbon-rich Arctic tundra soils! (2) Absence and presence of polygonal ground patterns is not directly related to carbon-richness of the soil (see e.g. Davis 2001). In fact, the non-orthogonal polygonal tundra patterns are mostly found on homogenous silty or sandy grounds, whereas the carbon-rich surfaces in my experience mostly show orthogonal polygonal patterns that differ from your site.

In fact, this all does not matter, it is simply a problematic first phrase (the one scientific writing is all about). Please rephrase and start your story in the direction where you actually go. In fact, only on page 12, line 8 my initial suspicion was resolved as you wrote "with its typically shallow organic horizon in the soil".

2. Abstract. I was confused by the your flux numbers. In principle, a negative sink is a source (page 1, line 8), but as an expert I guessed that you use the negative sign for net uptake and thus a sink of minus something is still a sink (not a source). OK, my recommendation is to put the number in parentheses to avoid the interpretation that it is a source. But the most confusing statement follows in the last line of the abstract: the text in lines 6 to 8 reads like: conventional calculation gives –46 gC m$^{-2}$, improved ogive optimization gives –82 gC m$^{-2}$ which is a **strengthening** of net uptake, but your

text on line 14 calls this "a weakening of the $CO_2$ sink" ... I assume you wanted the reader to read the abstract differently. Please reword and clarify. Maybe also define your sign convention in the abstract.

3. You strongly vote for the ogive optimization method. I am not perfectly in agreement with your argumentation, though. As I mentioned initially, it is good to lead the discussion, but some more critical assessment of this method is required, which should be reflected in revised wordings at serveral places.

Your example in Fig. 2b clearly shows gravity waves seen with the bands of lenticular clouds. Under such conditions it is challenging to filter out the waves (which should not be considered fluxes). In principle, such conditions should fail any stationarity test and one could thus think of other methods to filter out such conditions.

In the example given in Fig. 2a you basically truncate the turbulence cospectrum at 1/25 Hz, thus arguing that 25 seconds of measurements is enough to determine a half-hourly flux. This is in stark contrast to other concepts such as large eddy simulations where the generally accepted knowledge is used that it is the larger eddies—not the small ones—that are relevant for the turbulent fluxes between the surface and the atmosphere.

Long ago I had to deal with a similar issue with my first measurements over lakes (Eugster et al. 2003) and there I used the direction of the momentum flux as a filter criterion. However, some software compute the momentum flux in a way that loses the directional sign so that it is unclear in which direction the momentum flux actually pointed. In principle the momentum flux (averaged over 30 minutes) should point towards the ground surface, but my experience is that in cases as you show in Fig. 2 there might be an upward momentum flux in the high frequencies which would question your interpretation that these high frequencies are better to estimate the local $CO_2$ flux. This only holds if your momentum flux in the high frequency range is clearly downwards. To accept your interpretation I would need to see the cospectrum or ogive of the

horizontal windspeed time trace and the vertical windspeed time trace. The horizontal direction must be aligned with the flow so that $\overline{v'} = 0$ and $\overline{u} > 0$ m s$^{-1}$. Otherwise, if the turbulent momentum flux is in the wrong direction then your argument that the corresponding $CO_2$ flux must be from the local surface would be incorrect.

Maybe you have also measured a wind profile? If the peak wind speed near the surface is **below** your eddy covariance (EC) measurement height, then this would be a condition where the momentum flux measured by EC is upwards, not towards the ground. I must admit that filtering with momentum flux direction is very rigorous and in many cases may be overly picky, but I hope I could explain you why I am not really of the same opinion as you (page 13, lines 12–18): if momentum flux is upwards, then your EC system sees the inversion interface between the cold air on the surface and the warm air aloft (which is present if you have clouds as those shown in Fig. 2), not the ground interface. You may overcome this with a more critical rewording; your text on lines 13–14 does not really provide a realistic "speculation".

4. The limitations you list on page 2, lines 9–10 do not include the factor of self-heating if an open-path instrument is used. Later we see that you used a Licor 7200, but since your introduction is more general I recommend adding a statement here (many use open-path instruments in the Arctic due to power constraints). This is a factor that Baldocchi (2003) was not aware off, thus you should mention this after the citation.

5. There is confusion about your argument why you focus on 2014/2015 data and less on 2013: on page 3, line 3 you write: "were only recorded as wet molar densities and without the cell pressure necessary to convert them to dry mixing ratios". Is this a typo or did I misunderstand this statement? It is the $H_2O$ density measurement that is needed to convert from wet to dry mixing ratios. Temperature and cell pressure are only necessary to convert from densities to mixing ratios. A similar confusion is found on page 8, lines 1–2: "since they were wet rather than dry molar densities". Before you argued because of the mixing ratios, here one wonders why the ogive method should not work with wet molar densities if it would work with dry molar densities?

Please clarify these things. In principle you could use the Webb-Pearman-Leuning correction for your 2013 fluxes. Why did you not use this method to better profit from your interesting dataset?

**Detailed technical remarks**

1/10: use K instead of °C for temperature differences

3/1: add "flux" in EC $CO_2$ flux measurements

4/21: use "s" not "sec"

5/10: correlations between time series depend on the measurement interval; an $r > 0.9$ definitely does not hold for 10 Hz data, but may be seen with monthly data. Either specify which aggregation level you talk about, or simply remove this statement in parentheses. Giving the distance is an objective information that should be sufficient.

6/11–12: wording reflects some inconsistency in your statistical testing. I assume you used a t-test, but if you write "on average 10 cm larger thaw depth" then this implies a one-sided t-tests (testing for "greater than"). The wording on the line below ("this difference is not statistically significant") however is the wording for a two-sided test. Please rectify.

7/9: do not specify "$p < 10^{-12}$" since statistical models are not supposed to be accurate down to $p < 10^{-12}$. Normally for low values it is sufficient to indicate something on the order of $p < 0.0001$ or so.

8/13: use K instead of °C for temperature differences

[Figure]

**References**

Davis, T. N. (2001) *Permafrost: a guide to frozen ground in transition*. University of Alaska Press, Fairbanks, Alaska, ISBN 1-889963-19-4.

Eugster, W., Kling, G., Jonas, T., McFadden, J. P., Wüest, A., MacIntyre, S., Chapin III, F. S. (2003) $CO_2$ exchange between air and water in an arctic Alaskan and midlatitude Swiss lake: Importance of convective mixing. *J. Geophys. Res.*, **108**, 4362–4380.

---

## Referee Comment (RC2) · L. Kutzbach (Referee) · 24 Feb 2017

The manuscript of Pirk et al. presents interesting analyses of the spatial variability of topography and land-atmosphere fluxes of CO2 within a high-arctic polygonal tundra.

The small-scale spatial variability of topography was analyzed by photogrammetry of aerial photographs, which was used to produce a visual map and a digital elevation model. For an assessment of geomorphological changes of the polygonal tundra in the last decades, the new map was compared with historical aerial photographs. The study shows that no such geomorphological changes due to permafrost degradation could be detected at the high-arctic study site on Svalbard although the mean annual air temperatures on Svalbard have strongly increased in the last decades. This interesting

result suggests a rather strong resilience of polygonal tundra to climate warming.

The small-scale spatial variability of land-atmosphere fluxes of CO2 was analyzed by separating the flux time series in periods with either wind directions from a drier landscape sector or in periods with wind directions from a wetter landscape sector, and separately analyzing the respective flux controls and flux balances for the two different sectors. The conclusion of this part of the study is also scientifically interesting and relevant as it indicates that drying of polygonal tundra, which might happen in many polygonal tundra areas due to permafrost degradation, will lead to a decrease of the CO2 sink capacity of these tundra landscapes.

Furthermore, the authors aimed at a better understanding of "how the spatial heterogeneity and larger-scale disturbances affect eddy covariance flux estimates by investigating the spectral composition of the eddy covariance signal". For this objective, they apply the ogive optimization method, which was only recently introduced by Sievers et al. (2015). Generally, I find the application of this new method and its comparison to the conventional eddy covariance method presented by this manuscript highly valuable and of great relevance for the eddy covariance flux community. However, I think that the study does not provide enough evidence and deep-enough discussion to substantiate their claim that the ogive optimization method produces more trustworthy results than the conventional method. I discuss this in more depth in the list of specific comments below.

The language of the manuscript is clear and easy to follow. The figures are of high quality.

I recommend the manuscript of Pirk et al. for publication in Biogeosciences after major revisions considering my comments above and below.

Specific comments:

(1) Page 1: Title: I suggest weakening the rather strong and general statement in the

second part of the title: "large overestimations by the conventional eddy covariance method". I think that it is not clear enough at this point, which of the two methods – the conventional or the ogive optimization – delivers more trustworthy results. It is definitely an important finding of this study that the two methods lead to such strongly deviating results, but for a decision which method should be preferred, a better understanding of the atmospheric flow or experimental set-up effects potentially causing these biases would be needed. Furthermore, if the title suggests that the main message of the article is that the conventional eddy covariance method overestimates the CO2 uptake, the existing theoretical knowledge about eddy covariance measurements over heterogeneous landscapes and complex terrain must be more extensively reflected both in the introduction and the discussion. If the main message of the article is on the biases of the eddy covariance method, it is not enough to just refer to the work of Sievers et al. (2015). Then, the authors have to discuss their findings in the light of the extensive work on eddy covariance measurements over heterogeneous landscapes and in complex terrain (e.g., Mahrt et al. (1994), Finnigan et al. (2003), Inagaki et al. (2006), Aubinet et al. (2010), and others) in the current manuscript.

(2) Page 2, lines 21-22: The paper of Kutzbach et al. (2007) reports an annual net ecosystem CO2 exchange (NEE) of $-71$ g CO2 m$-2$, which equals to about 19 g C m-2, for polygonal tundra (Kutzbach et al., 2007). Please correct this.

(3) Page 3, lines 7-9: I think that it would be important to more thoroughly describe and discuss the patterns of prevailing wind directions and the microclimatic situation in general. The investigation site is located in a valley surrounded by rather high mountains, and it is near to the sea (fjord). Therefore, sea and land breezes, katabatic or anabatic winds as well as gravity waves may have important effects on the air movements analyzed by the eddy covariance system. This could be relevant for the discussion of the observed frequency mismatches in the co-spectra and ogives, respectively.

(4) Page 3, lines 12-14: Please give here more information on the soil properties in this polygonal tundra. In particular, organic carbon contents in the different soils of

polygonal tundra would be of interest. Spatial variability of soil organic matter contents is likely pronounced in the polygonal tundra (Zubrzycki et al., 2013).

(5) Page 3, lines 14-16: Please give more detailed information about the vegetation composition within the polygonal tundra. How does vegetation differ between low-center polygons of different degradation/drainage conditions? Please give information on (approximate) ground coverages of shrubs, sedges and mosses at polygon rims and polygon centers of different water levels. The coverage of mosses is of high interest since they can start photosynthesizing directly after snowmelt (or even earlier) (Oechel , 1976, Tieszen et al., 1980). When discussion the early $CO_2$ sink function suggested by the conventional eddy covariance method, coverage of mosses is of interest.

(6) Page 3, line 23: This sentence is confusing. You need the pressure and temperature inside the cell to convert from molar densities to mixing ratios. You need water vapor measurements to convert from mole fractions (referred to wet air) to mixing ratios referred to dry air. Please write this in a clearer way.

(7) Page 6, lines 4ff: How did the footprint extents differ before, during and after snowmelt? The snow cover could have a significant effect on footprint extents due to its lower roughness length. Could this affect the flux co-spectra during snowmelt?

(8) Page 6, lines 13ff: When considering the pronounced spatial variability within the footprints of the eddy covariance measurements, I wonder how much you can be sure that the frequency mismatches are due to local and non-local flux contributions. Could this mismatch also be caused by flux heterogeneity within the (local) footprint? The position of the flux tower appears to be at a drier patch compared to the surroundings in the studied polygonal tundra. When moving from the tower in both main prevailing wind directions, the first wet polygons are found some 30 m to 50 m away from the tower. Could it be possible that the observed frequency mismatches (commonly sign of co-variance different for eddies larger than about 30 m than for eddies smaller than 30 m) are due to positive $CO_2$ fluxes from the drier polygons near the tower (reflected better

in the high frequencies) and negative CO2 fluxes at the wetter tundra at larger distance from the tower (reflected in the low frequencies)? If wetter tundra has more mosses, this could lead to earlier negative fluxes than at drier sites with less mosses since mosses can start photosynthesizing directly after snowmelt (or even earlier) (Oechel , 1976, Tieszen et al., 1980). Since the strongest frequency mismatches were observed during the snowmelt period, it would be also very interesting to have more information on the snow distribution: Was there the same snow coverage near the flux tower in the drier polygons than further away (30-50 m) in the wetter polygons?

(9) Page 6, line 30: What do you mean with "better performance"? How did you assess "performance"? I think that CO2 uptake during the snowmelt period (as it is illustrated in in Figure 2b) would not be as implausible as suggested by the authors since mosses can start photosynthesizing directly after snowmelt (or even earlier, see above).

(10) Page 7, line 6: What is exactly meant by "combined footprint"? Just using the original eddy covariance flux time series without separating periods of different wind directions? Or have you applied some sort of spatial weighing of the contributions of wetter and drier polygonal tundra to the whole are of interest? If you do the former, then the CO2 balances for the "combined footprint" would depend to a large degree on the frequency distribution of wind directions.

(11) Page 7, lines 9-10; Page 8, lines 1-2: It does not become clear why you can calculate eddy covariance fluxes without having mixing ratios referred to dry air by using the conventional EddyPro method but not by using the ogive optimization method. Couldn't you apply the classic WPL approach (Webb et al. (1980) as refined by Ibrom et al. (2007)) to fluxes calculated by both methods?

(12) Page 12, lines 7-8: The observed annual CO2 uptake appears indeed very large. However, I think that such an uptake is well possible. For example, high CO2 uptake was also observed at coastal wet sedge tundra near Barrow, Alaska, by Harazono et al. (2003).

References: Aubinet, M., Feigenwinter, C., Heinesch, B., Bernhofer, C., Canepa, E., Lindroth, A., ... & Van Gorsel, E. (2010). Direct advection measurements do not help to solve the night-time CO 2 closure problem: evidence from three different forests. Agricultural and forest meteorology, 150(5), 655-664. Finnigan, J. J., Clement, R., Malhi, Y., Leuning, R., & Cleugh, H. A. (2003). A re-evaluation of long-term flux measurement techniques part I: averaging and coordinate rotation. Boundary-Layer Meteorology, 107(1), 1-48. Harazono, Y., Mano, M., Miyata, A., Zulueta, R. C., & Oechel, W. C. (2003). Inter‐annual carbon dioxide uptake of a wet sedge tundra ecosystem in the Arctic. Tellus B, 55(2), 215-231. Ibrom, A., E. Dellwik, S. E. Larse, and K. Pilegaard. (2007). On the use of the Webb-Pearman-Leuning theory for closed-path eddy correlation measurements, Tellus Series B-Chemical and Physical Meteorology, 59:937-946. Inagaki, A., Letzel, M. O., Raasch, S., & KANDA, M. (2006). Impact of surface heterogeneity on energy imbalance: a study using LES. Journal of the Meteorological Society of Japan. Ser. II, 84(1), 187-198. Mahrt, L., Sun, J., Vickers, D., MacPherson, J. I., Pederson, J. R., & Desjardins, R. L. (1994). Observations of fluxes and inland breezes over a heterogeneous surface. Journal of the atmospheric sciences, 51(17), 2484-2499. Oechel WC. 1976. Seasonal patterns of temperature response of CO2 flux and acclimation in arctic mosses growing in situ. Photosynthetica 10: 447-456. Tieszen LL, Miller PC, Oechel WC. 1980. Photosynthesis. In: Brown et al (eds) An Arctic Ecosystem: the Coastal Tundra at Barrow, Alaska. Dowden, Hutchinson & Ross Inc, Stroudsburg, PA, USA. pp 102-139. Webb, E. K., G. I. Pearman, and R. Leuning (1980). Correction of flux measurements for density effects due to heat and water vapor transfer. Quarterly Journal of the Royal Meteorological Society, 106: 85–100. Zubrzycki, S., Kutzbach, L., Grosse, G., Desyatkin, A., and Pfeiffer, E.-M. (2013): Organic carbon and total nitrogen stocks in soils of the Lena River Delta, Biogeosciences, 10, 3507-3524, doi:10.5194/bg-10-3507-2013, 2013.

---

## Author Comment (AC1) · 24 Mar 2017

*Comment: The authors measured net CO2 fluxes over polygonal tundra in the high arctic on Svalbard. They present an interesting and well written manuscript, but seem to interpret some aspects differently than this reviewer. I think it is important to exchange ideas and opinions between authors and reviewers and that it is positive to have new ideas presented even if not everyone agrees. However, my main points given below is that the language should reflect this in a somewhat clearer way so that the uninitiated reader does not misinterpret the universality of some statements.*
*Having said this, I must admit that I learned a lot reading this manuscript and fully support its publication after careful revisions.*
*Besides an important methodological aspect of how to compute defensible annual flux sums, a key statement of the paper is that all the detailed image analyses starting with pictures taken in 1948 cannot confirm a rapid degradation of this polygonal tundra, but rather support the view that this landscape has been quite stable over the last seven decades.*

**Reply:** We thank Professor Eugster for his thorough review of our manuscript and his helpful feedback. We carefully considered each of his comments, paying special attention to the potentially exaggerated universality of some of our statements.

**Main critique**

*Comment: 1. Your introduction completely misguided me in the wrong direction as your paper starts with the phrase "Carbon-rich Arctic tundra soils are often covered with polygonal ground patterns created by sub-surface ice wedges." – (1) Your paper is not addressing carbon-rich Arctic tundra soils! (2) Absence and presence of polygonal ground patterns is not directly related to carbon-richness of the soil (see e.g. Davis 2001). In fact, the non-orthogonal polygonal tundra patterns are mostly found on homogenous silty or sandy grounds, whereas the carbon-rich surfaces in my experience mostly show orthogonal polygonal patterns that differ from your site.*
*In fact, this all does not matter, it is simply a problematic first phrase (the one scientific writing is all about). Please rephrase and start your story in the direction where you actually go. In fact, only on page 12, line 8 my initial suspicion was resolved as you wrote "with its typically shallow organic horizon in the soil".*

**Reply:** The top 100 cm of the soil in the EC footprint at the Adventdalen site contain about 30 kg SOC m-2 on average (personal communication with Peter Kuhry). In comparison with other permafrost-underlain well-developed soils, which can contain >100 kg SOC m-2 in the top 100 cm (see e.g. Hugelius et al. 2014), Adventdalen is not extremely carbon-rich.  So we understand the confusion created by our first sentence and propose to resolve this misunderstanding by removing "carbon-rich" in the first sentence, so that it would read: "Arctic tundra is often covered with polygonal ground patterns created by sub-surface ice wedges."

**Comment:** *2. Abstract. I was confused by the your flux numbers. In principle, a negative sink is a source (page 1, line 8), but as an expert I guessed that you use the negative sign for net uptake and thus a sink of minus something is still a sink (not a source). OK, my recommendation is to put the number in parentheses to avoid the interpretation that it is a source. But the most confusing statement follows in the last line of the abstract: the text in lines 6 to 8 reads like: conventional calculation gives –46 gC m–2, improved ogive optimization gives –82 gC m–2 which is a strengthening of net uptake, but your text on line 14 calls this "a weakening of the CO2 sink" . . . I assume you wanted the reader to read the abstract differently. Please reword and clarify. Maybe also define your sign convention in the abstract.*

**Reply:** We acknowledge that these formulations could be misinterpreted and need clarification. So we propose to put these numbers in parentheses and define the sign convention at the first occurrence. The sentences 5 and 6 of the abstract would then read: "Non-local (low-frequency) flux contributions were especially pronounced during snowmelt and introduced a large bias of -46 gC m-2 to the annual $CO_2$ budget in conventional methods (minus-sign indicating a higher uptake by the ecosystem). Our improved flux calculations with the ogive optimization method indicated that the site was a strong sink for $CO_2$ in 2015 (-82 gC m-2) and due to differences in light-use efficiency, wetter areas with low-centered polygons sequestered 47% more $CO_2$ than drier areas with flat-centered polygons."

**Comment:** *3. You strongly vote for the ogive optimization method. I am not perfectly in agreement with your argumentation, though. As I mentioned initially, it is good to lead the discussion, but some more critical assessment of this method is required, which should be reflected in revised wordings at several places. Your example in Fig. 2b clearly shows gravity waves seen with the bands of lenticular clouds. Under such conditions it is challenging to filter out the waves (which should not be considered fluxes). In principle, such conditions should fail any stationarity test and one could thus think of other methods to filter out such conditions.*

**Reply:** Firstly, the picture shown in Fig. 2b was not meant to represent the exact same 30-min period used in Fig. 2a, but only approximately the time of year during snowmelt. Using the time stamp of this photo, we now derived the matching EC ogives, which show the same features as the time period used in the original manuscript:

[Figure]

Admittedly, the data quality (QC) is worse and the ogive density map does indicate a degree of non-stationarity. So Prof. Eugster rightly points out that the period shown in Fig. 2b is not ideal for EC flux measurements with any method. However, this gravity wave event seemed to be rather short-lived because no lenticular clouds could be seen on photos taken 3 hours earlier:

[Figure]

In fact, such lenticular clouds have otherwise never been observed during our site visits so the original picture shown in Fig. 2b is not very representative. To resolve this issue and minimize the potential for misunderstandings, we propose to exchange Fig. 2b in the revised manuscript with this picture, and indicate its time stamp in the figure caption (27 May 2015, 12:00 LT):

[Figure]

**Comment:** *In the example given in Fig. 2a you basically truncate the turbulence cospectrum at 1/25 Hz, thus arguing that 25 seconds of measurements is enough to determine a half-hourly flux. This is in stark contrast to other concepts such as large eddy simulations where the generally accepted knowledge is used that it is the larger eddies—not the small ones—that are relevant for the turbulent fluxes between the surface and the atmosphere.*

**Reply:** The shortest dataset evaluated by the ogive optimization software is 10 min and the longest one is 60 min, thus covering a wider range of turbulence scales than conventional fixed-averaging methods. The example in Fig. 2a shows that all relevant flux contributions are carried by turbulence with a scale shorter than 25 sec. While that is interesting, we don't think it is particularly uncommon for EC measurements to have quite small flux contributions below this frequency. However, we definitely don't want to say that 25 sec are in general enough to determine a 30 min flux. To clarify this point, we propose to add the following sentence to the "Results" section describing Fig. 2a: "The ogive optimization model indicates that all relevant flux contributions are carried by turbulence with a scale shorter than about 25 sec in this example (which does, however, not mean that 25 sec are in general enough to determine a 30 min flux)."

**Comment:** *Long ago I had to deal with a similar issue with my first measurements over lakes (Eugster et al. 2003) and there I used the direction of the momentum flux as a filter criterion. However, some software compute the momentum flux in a way that loses the directional sign so that it is unclear in which direction the momentum flux actually pointed. In principle the momentum flux (averaged over 30 minutes) should point towards the ground surface, but my experience is that in cases as you show in Fig. 2 there might be an upward momentum flux in the high frequencies which would question your interpretation that these high frequencies are better to estimate the local CO2 flux. This only holds if your momentum flux in the high frequency range is clearly downwards. To accept your interpretation I would need to see the cospectrum or ogive of the horizontal windspeed time trace and the vertical windspeed time trace. The horizontal direction must be aligned with the flow so that $v' = 0$ and $u > 0$ ms$-1$. Otherwise, if the turbulent momentum flux is in the wrong direction then your argument that the corresponding CO2 flux must be from the local surface would be incorrect.*
*Maybe you have also measured a wind profile? If the peak wind speed near the surface is below your eddy covariance (EC) measurement height, then this would be a condition where the momentum flux measured by EC is upwards, not towards the ground. I must admit that filtering with momentum flux direction is very rigorous and in many cases may be overly picky, but I hope I could explain you why I am not really of the same opinion as you (page 13, lines 12–18): if momentum flux is upwards, then your EC system sees the inversion interface between the cold air on the surface and the warm air aloft (which is present if you have clouds as those shown in Fig. 2), not the ground interface. You may overcome this with a more critical rewording; your text on lines 13–14 does not really provide a realistic "speculation".*

**Reply:** The ogive optimization software checks that the momentum flux is negative (towards the ground surface) in the mid-frequency range as one of the quality checks. While this test is described in the original publication of the method, we acknowledge that it would be relevant to briefly mention this test in the "Methods/Data processing" section of our manuscript. We therefore propose to add this sentence: "Ogive optimization furthermore only accepts periods with a negative momentum flux (i.e. directed toward the ground surface) in the mid-frequency range ($10^{-1.5}$ Hz)."

We don't have vertical wind profile measurements, but we specifically checked the momentum flux ogives for the period given in Fig. 2a:

[Figure]

These ogives indicate a well-behaved and downwards-directed momentum flux, so we are confident that our arguments hold.

Regarding our speculation of a vertical $CO_2$ layering (page 13, line 13ff), we fail to see why it would be so unrealistic, and since it's clearly indicated as a speculation we cannot really express it more carefully. Still, the possible stability layering suggested by Prof. Eugster could also be at work, so we propose to broaden this speculation a little bit to also mention possible layers of different atmospheric stability, so that the revised sentence would read: "One might speculate that the systematic occurrences of bi-directional fluxes are due to an atmospheric layering where low and high-frequency eddies circle through air masses with different atmospheric stability and $CO_2$ concentrations"

***Comment:*** *4. The limitations you list on page 2, lines 9–10 do not include the factor of self-heating if an open-path instrument is used. Later we see that you used a Licor 7200, but since your introduction is more general I recommend adding a statement here (many use open-path instruments in the Arctic due to power constraints). This is a factor that Baldocchi (2003) was not aware off, thus you should mention this after the citation.*

**Reply:** OK, we propose to add the following sentence here: "Also, when open-path gas analyzers are used, a bias may be introduced due to surface heating of the instrument itself (Burba et al., 2008)."

***Comment:*** *5. There is confusion about your argument why you focus on 2014/2015 data and less on 2013: on page 3, line 3 you write: "were only recorded as wet*

*molar densities and without the cell pressure necessary to convert them to dry mixing ratios". Is this a typo or did I misunderstand this statement? It is the H2O density measurement that is needed to convert from wet to dry mixing ratios. Temperature and cell pressure are only necessary to convert from densities to mixing ratios. A similar confusion is found on page 8, lines 1–2: "since they were wet rather than dry molar densities". Before you argued because of the mixing ratios, here one wonders why the ogive method should not work with wet molar densities if it would work with dry molar densities?*
*Please clarify these things. In principle you could use the Webb-Pearman-Leuning correction for your 2013 fluxes. Why did you not use this method to better profit from your interesting dataset?*

**Reply:** We acknowledge that these statements are somewhat inconsistent and misleading. The problem with the 2013 data is that a sample-by-sample conversion of recorded densities to mixing ratios would require the cell pressure, which hasn't been recorded. And while EddyPro can use the WPL correction to calculate corrected fluxes, this feature is not implemented in the current version of the ogive optimization software. We propose to clarify this issue in the "Methods/Data processing" section, writing: "We mainly focused on data collected between September 2014 and December 2015, when data quality and coverage was highest. $CO_2$ concentrations collected in 2013 were only recorded as molar densities and without the cell pressure necessary for a sample-by-sample conversion to mixing ratios according to the Webb-Pearman-Leuning correction proposed by Sahlee et al. (2008), which is currently the only option implemented in the ogive optimization software. Hence, we only report 2013's fluxes from EddyPro as supplementary support for our findings."

The misleading sentence on page 3 line 23 would be removed. The sentence on page 8 lines 1-2 could then be simplified, reading:
"In 2013, EddyPro calculations yielded a smaller total annual $CO_2$ balance of -79 gC m-2 (see Fig. S2d in the supporting information), whereas ogive optimization fluxes could not be calculated from 2013's raw $CO_2$ measurements (cf. section 2.3)."

**Detailed technical remarks**

**Comment:** *1/10: use K instead of ◦C for temperature differences*
**Reply:** OK

**Comment:** *3/1: add "flux" in EC CO2 flux measurements*
**Reply:** OK

**Comment:** *4/21: use "s" not "sec"*
**Reply:** OK, we changed this throughout the entire manuscript.

**Comment:** *5/10: correlations between time series depend on the measurement interval; an r > 0.9 definitely does not hold for 10 Hz data, but may be seen with*

*monthly data. Either specify which aggregation level you talk about, or simply remove this statement in parentheses. Giving the distance is an objective information that should be sufficient.*
**Reply:** OK, we removed the statement in parentheses.

*Comment: 6/11–12: wording reflects some inconsistency in your statistical testing. I assume you used a t-test, but if you write "on average 10 cm larger thaw depth" then this implies a one-sided t-tests (testing for "greater than"). The wording on the line below ("this difference is not statistically significant") however is the wording for a two-sided test. Please rectify.*
**Reply:** OK, we propose to clarify that we used a two-sided t-test by changing this part to: "The thaw depth at the centers of the polygons around the EC tower was 66cm +/-9cm (mean +/-standard deviation, sample size N=30) by the end of August. Based on the polygons in the 50% EC footprint, the drier ESE fetch area featured a thaw depth of 69 cm +/-8 cm (N=4) while the wetter NW featured 79 cm +/-4 cm (N=4), which is not a statistically significant difference (p=0.10)."

*Comment: 7/9: do not specify "p < 10−12" since statistical models are not supposed to be accurate down to p < 10−12. Normally for low values it is sufficient to indicate something on the order of p < 0.0001 or so.*
**Reply:** OK

*Comment: 8/13: use K instead of ◦C for temperature differences*
**Reply:** OK

References:

Hugelius, Gustaf, et al. "Estimated stocks of circumpolar permafrost carbon with quantified uncertainty ranges and identified data gaps." Biogeosciences 11.23 (2014): 6573-6593.

Burba, George G., et al. "Addressing the influence of instrument surface heat exchange on the measurements of $CO_2$ flux from open‐path gas analyzers." *Global Change Biology* 14.8 (2008): 1854-1876.

---

## Author Comment (AC2) · 24 Mar 2017

*Comment: The manuscript of Pirk et al. presents interesting analyses of the spatial variability of topography and land-atmosphere fluxes of CO2 within a high-arctic polygonal tundra.*

*The small-scale spatial variability of topography was analyzed by photogrammetry of aerial photographs, which was used to produce a visual map and a digital elevation model. For an assessment of geomorphological changes of the polygonal tundra in the last decades, the new map was compared with historical aerial photographs. The study shows that no such geomorphological changes due to permafrost degradation could be detected at the high-arctic study site on Svalbard although the mean annual air temperatures on Svalbard have strongly increased in the last decades. This interesting result suggests a rather strong resilience of polygonal tundra to climate warming.*

*The small-scale spatial variability of land-atmosphere fluxes of CO2 was analyzed by separating the flux time series in periods with either wind directions from a drier landscape sector or in periods with wind directions from a wetter landscape sector, and separately analyzing the respective flux controls and flux balances for the two different sectors. The conclusion of this part of the study is also scientifically interesting and relevant as it indicates that drying of polygonal tundra, which might happen in many polygonal tundra areas due to permafrost degradation, will lead to a decrease of the CO2 sink capacity of these tundra landscapes.*

*Furthermore, the authors aimed at a better understanding of "how the spatial heterogeneity and larger-scale disturbances affect eddy covariance flux estimates by investigating the spectral composition of the eddy covariance signal". For this objective, they apply the ogive optimization method, which was only recently introduced by Sievers et al. (2015). Generally, I find the application of this new method and its comparison to the conventional eddy covariance method presented by this manuscript highly valuable and of great relevance for the eddy covariance flux community. However, I think that the study does not provide enough evidence and deep-enough discussion to substantiate their claim that the ogive optimization method produces more trustworthy results than the conventional method. I discuss this in more depth in the list of specific comments below.*

*The language of the manuscript is clear and easy to follow. The figures are of high quality.*

*I recommend the manuscript of Pirk et al. for publication in Biogeosciences after major revisions considering my comments above and below.*

**Reply:** We thank Prof. Lars Kutzbach for his thorough review and his comments to further improve our manuscript.

**Specific comments:**

*Comment: (1) Page 1: Title: I suggest weakening the rather strong and general statement in the second part of the title: "large overestimations by the conventional eddy covariance method". I think that it is not clear enough at this point, which of the two methods – the conventional or the ogive optimization – delivers more trustworthy results. It is definitely an important finding of this study that the two methods lead to such strongly deviating results, but for a decision which method*

*should be preferred, a better understanding of the atmospheric flow or experimental set-up effects potentially causing these biases would be needed. Furthermore, if the title suggests that the main message of the article is that the conventional eddy covariance method overestimates the CO2 uptake, the existing theoretical knowledge about eddy covariance measurements over heterogeneous landscapes and complex terrain must be more extensively reflected both in the introduction and the discussion. If the main message of the article is on the biases of the eddy covariance method, it is not enough to just refer to the work of Sievers et al. (2015). Then, the authors have to discuss their findings in the light of the extensive work on eddy covariance measurements over heterogeneous landscapes and in complex terrain (e.g., Mahrt et al. (1994), Finnigan et al. (2003), Inagaki et al. (2006), Aubinet et al. (2010), and others) in the current manuscript.*

**Reply:** OK, we propose to weaken the statement in the second part of the title to take some weight off the eddy covariance calculations. So the revised title would be: "Spatial variability of CO2 uptake in polygonal tundra – assessing low-frequency disturbances in eddy covariance flux estimates"

We furthermore propose to extend the introduction including the mentioned literature by adding the sentences: "Large-scale surface heterogeneity has been observed and simulated to induce thermal circulations on the mesoscale that can impede the turbulent flux estimation (Mahrt et al., 1994; Inagaki et al., 2006), while complex terrain may lead to horizontal advection of gases and thereby biased flux estimations (Finnigan et al., 2003; Aubinet et al., 2010). Finnigan et al. (2003) showed that the averaging operation and coordinate rotation commonly applied in EC flux calculations can lead to co-spectral distortions and a loss of flux."

In our discussion, we propose to make specific mention of the CO2 co-spectra given in Figures 15 and 16 of Finnigan et al. (2003), which show an indication of a similar frequency mismatch below 10^-3 Hz:

[Figure]

*Figure 15.* Ensemble averaged scalar cospectra for the period 0800–1200 EST from 9 days of Tumbarumba data. The cospectra are plotted in area preserving form. Each plot consists of four curves corresponding to averaging and rotation periods of 15 min, 1 h, 2 h and 4 h. The curves extend to successively lower frequency as the averaging period increases. (a) Sensible heat, H. (b) Latent heat, *LE* from the closed-path Licor instrument. No high frequency correction has been applied. (c) Carbon dioxide flux, $F_C$ from the closed-path Licor instrument. No high frequency correction has been applied.

The fact that the low frequency shift is here only observed for CO2 may have something to do with the low covariance of CO2 observations, relative to sensible and latent heat fluxes.

**Comment:** *(2) Page 2, lines 21-22: The paper of Kutzbach et al. (2007) reports an annual net ecosystem CO2 exchange (NEE) of −71 g CO2 m−2, which equals to about 19 g C m-2, for polygonal tundra (Kutzbach et al., 2007). Please correct this.*

**Reply:** OK, we corrected this.

**Comment:** *(3) Page 3, lines 7-9: I think that it would be important to more thoroughly describe and discuss the patterns of prevailing wind directions and the microclimatic situation in general. The investigation site is located in a valley surrounded by rather high mountains, and it is near to the sea (fjord). Therefore,*

*sea and land breezes, katabatic or anabatic winds as well as gravity waves may have important effects on the air movements analyzed by the eddy covariance system. This could be relevant for the discussion of the observed frequency mismatches in the co-spectra and ogives, respectively.*

**Reply:** OK, we propose to extend the site description with the following sentences: "The surrounding mountains feature plateaus of around 450 m a.s.l., as well as peaks and ridges of up to 1000 m a.s.l, which are still partly glaciated. Wind directions are generally oriented along the valley, with dominating easterlies in wintertime (coming from inland Spitsbergen), and an approximately even distribution of easterlies and westerlies in summertime (westerlies coming from the fjord). Long-term statistics indicate that wind speeds in Adventdalen are below 5 m s-1 for about 70% of the year (and below 10 m s-1 for about 97%), with a most frequent wind speed of about 3 m s-1."

*Comment: (4) Page 3, lines 12-14: Please give here more information on the soil properties in this polygonal tundra. In particular, organic carbon contents in the different soils of polygonal tundra would be of interest. Spatial variability of soil organic matter contents is likely pronounced in the polygonal tundra (Zubrzycki et al., 2013).*

**Reply:** OK, we propose to also extend this part of the site description with what has been reported by other studies: "The measurement site is located on a river terrace on the flat part of a large alluvial fan, where the ground is patterned by ice-wedge polygons. These coarse alluvial deposits are covered with a few ten centimeters of organic material and fine-grained eolian deposits (loess), which typically stem from wind erosion in the braided riverbed when it dries out in autumn (Bryant et al. (1982), Oliva et al. (2014)). The site's soil organic carbon content in the uppermost 100 cm soil is about 30 kgC m-2 (personal communication with Peter Kuhry)."

*Comment: (5) Page 3, lines 14-16: Please give more detailed information about the vegetation composition within the polygonal tundra. How does vegetation differ between low- center polygons of different degradation/drainage conditions? Please give information on (approximate) ground coverages of shrubs, sedges and mosses at polygon rims and polygon centers of different water levels. The coverage of mosses is of high interest since they can start photosynthesizing directly after snowmelt (or even earlier) (Oechel , 1976, Tieszen et al., 1980). When discussion the early CO2 sink function suggested by the conventional eddy covariance method, coverage of mosses is of interest.*

**Reply:** Due to the patterned microtopography, there is considerable variability in the vegetation cover. A dedicated vegetation analysis has not been conducted, so we can unfortunately not estimate the overall moss cover. Our general assessment is that the three vegetation layers (shrubs, sedges, mosses) clearly overlap in most areas. Drier areas are dominated by shrubs and sedges, while wetter areas are dominated by sedges and mosses. So we can assume sufficient

moss coverage for the discussion of the early onset of net CO2 uptake indicated by the conventional method (details given in reply to comment 8 below).
To give some more information about the specifically relevant moss cover, we propose to add these sentences to the site description: "The moss cover is sparse in drier polygons where shrubs dominate the vegetation community, while the wetter areas at local depressions feature an almost continuous moss cover. Within individual polygons the moss coverage typically increases from the drier rim to the wetter center."

**Comment:** *(6) Page 3, line 23: This sentence is confusing. You need the pressure and temperature inside the cell to convert from molar densities to mixing ratios. You need water vapor measurements to convert from mole fractions (referred to wet air) to mixing ratios referred to dry air. Please write this in a clearer way.*

**Reply:** We acknowledge that this sentence needs clarification (as also pointed out in comment 5 by reviewer #1). We propose to change it to: "CO2 concentrations collected in 2013 were only recorded as molar densities and without the cell pressure necessary for a sample-by-sample conversion to mixing ratios according to the Webb-Pearman-Leuning correction proposed by Sahlee et al. (2008), which is currently the only option implemented in the ogive optimization software. Hence, we only report 2013's fluxes from EddyPro as supplementary support for our findings"

**Comment:** *(7) Page 6, lines 4ff: How did the footprint extents differ before, during and after snowmelt? The snow cover could have a significant effect on footprint extents due to its lower roughness length. Could this affect the flux co-spectra during snowmelt?*

**Reply:** In our footprint estimation we kept the roughness length constant at 1 cm throughout the year. This value might be slightly too high for snow (which is typically assigned 0.5 cm) and maybe slightly too low for open tundra vegetation (typically assigned 3 cm). We have however not undertaken dedicated efforts to quantify this parameter more precisely. Therefore, we might overestimate the footprint extent a little bit during the snow-free season, and underestimate it a little during snow-covered conditions.
The surface roughness could indeed affect the flux co-spectra -- in principle at any time of year. One conceivable mechanism is that a greater roughness length could break down larger turbulence into smaller turbulence, thus shifting some of the co-spectrum toward higher frequencies. We propose to add this consideration to our discussion, stating: "Snowmelt also entails a change in the typical surface roughness length, which is slightly smaller for snow than open tundra vegetation. A greater roughness could break down larger turbulence into smaller turbulence, thus shifting some of the flux co-spectrum toward higher frequencies. However, such spectral shifts would be no problem for the functionality of the used flux calculation schemes and cannot readily explain bi-directional fluxes even if the surface roughness is spatially heterogeneous."

**Comment:** *(8) Page 6, lines 13ff: When considering the pronounced spatial variability within the footprints of the eddy covariance measurements, I wonder how much you can be sure that the frequency mismatches are due to local and non-local flux contributions. Could this mismatch also be caused by flux heterogeneity within the (local) footprint? The position of the flux tower appears to be at a drier patch compared to the surroundings in the studied polygonal tundra. When moving from the tower in both main prevailing wind directions, the first wet polygons are found some 30 m to 50 m away from the tower. Could it be possible that the observed frequency mismatches (commonly sign of covariance different for eddies larger than about 30 m than for eddies smaller than 30 m) are due to positive CO2 fluxes from the drier polygons near the tower (reflected better in the high frequencies) and negative CO2 fluxes at the wetter tundra at larger distance from the tower (reflected in the low frequencies)? If wetter tundra has more mosses, this could lead to earlier negative fluxes than at drier sites with less mosses since mosses can start photosynthesizing directly after snowmelt (or even earlier) (Oechel , 1976, Tieszen et al., 1980). Since the strongest frequency mismatches were observed during the snowmelt period, it would be also very interesting to have more information on the snow distribution: Was there the same snow coverage near the flux tower in the drier polygons than further away (30-50 m) in the wetter polygons?*

**Reply:** When we first noticed the systematic frequency mismatches during snowmelt we were thinking along exactly the same lines as Prof. Kutzbach describes here, namely that different frequencies represent different areas, of which some are CO2 sources and others CO2 sinks. However, due to the comparably large scale at which is mismatch occurs (around 25 sec, corresponding to more than 30 meter), we largely discarded this explanation again. We don't have a detailed vegetation map or a good estimation of the snow coverage throughout the snowmelt period, but we generally estimate the patchiness during snowmelt to be smaller than the size of the eddies corresponding to the lower frequencies.

Moreover, we observed the frequency mismatches from both wind directions, and since there are likely less mosses in the east than the west, the moss mechanism for this mismatch becomes even less likely.

However, we cannot fully exclude the mechanism outlined by Prof. Kutzbach, so we propose to add the following sentences to our discussion: "However, we cannot fully exclude that the frequency mismatches are caused by flux heterogeneity within the local footprint. It could be possible that the drier areas near the EC tower (reflected better in the high frequencies) are net CO2 sources, while wetter areas at larger distances from the tower (reflected in the low frequencies) are net CO2 sinks. A heterogeneous vegetation composition might cause such flux heterogeneity during snowmelt, because unlike shrubs and sedges, mosses have photosynthetically active tissue that may overwinter so that they can start photosynthesizing at low rates already during snowmelt (Oechel et al. (1976), Tieszen et al. (1980)). While some degree flux heterogeneity is certainly present at any time of year, its effect might be too small to explain the large frequency mismatches observed particularly during snowmelt. Nevertheless, our observations might incentivize future studies to investigate the frequency dependency of EC footprints."

**Comment:** *(9) Page 6, line 30: What do you mean with "better performance"? How did you assess "performance"? I think that CO2 uptake during the snowmelt period (as it is illustrated in in Figure 2b) would not be as implausible as suggested by the authors since mosses can start photosynthesizing directly after snowmelt (or even earlier, see above).*

**Reply:** We acknowledge that "better performance" is not the best formulation here and propose to change this sentence to: "Since the frequency mismatches cannot be resolved in conventional calculations, we focus on the NEE fluxes calculated by the ogive optimization method for the ecosystem characterization."

**Comment:** *(10) Page 7, line 6: What is exactly meant by "combined footprint"? Just using the original eddy covariance flux time series without separating periods of different wind directions? Or have you applied some sort of spatial weighing of the contributions of wetter and drier polygonal tundra to the whole are of interest? If you do the former, then the CO2 balances for the "combined footprint" would depend to a large degree on the frequency distribution of wind directions.*

**Reply:** Yes, we simply mean using all fluxes without separating wind directions, and we agree that the derived balances will depend on the occurrence of the two wind directions. We still chose to report these CO2 balances because this seems to be quite common for EC studies. To clarify the meaning of "combined footprint" in this case, we propose to change this sentence to: "The annual balance of the combined footprint (using all fluxes without separating periods of different wind directions) was -82 gC m-2 in 2015."

**Comment:** *(11) Page 7, lines 9-10; Page 8, lines 1-2: It does not become clear why you can calculate eddy covariance fluxes without having mixing ratios referred to dry air by using the conventional EddyPro method but not by using the ogive optimization method. Couldn't you apply the classic WPL approach (Webb et al. (1980) as refined by Ibrom et al. (2007)) to fluxes calculated by both methods?*

**Reply:** We acknowledge that our formulations of this issue weren't clear enough (as also noted by reviewer #1). While it is in principle possible to use the WPL approach in ogive optimization, it is not implemented in the current version of this software. So this task remains to be solved in future studies.
We propose to clarify this part of the text, as noted in our reply to comment 5 of reviewer #1.

**Comment:** *(12) Page 12, lines 7-8: The observed annual CO2 uptake appears indeed very large. However, I think that such an uptake is well possible. For example, high CO2 uptake was also observed at coastal wet sedge tundra near Barrow, Alaska, by Harazono et al. (2003).*

**Reply:** We thank the reviewer for pointing us toward this publication by Harazono et al. This paper only reports budgets from spring to autumn, i.e. not the full annual $CO_2$ budget including possible wintertime release of $CO_2$. Still, we propose to mention this large growing season uptake in our discussion by adding the following sentence: "The summertime $CO_2$ sink of Adventdalen is comparable to that of a coastal wet sedge tundra ecosystem at Barrow, Alaska (-105 to -162 gC m-2) (Harazono et al. (2003))."

---

## Author Response (AR2)

**Reply to reviewer #1**

We thank Prof. Kutzbach for reviewing our manuscript again and accepting the revised version of it.

**Reply to reviewer #2**

*Comment: The authors did a great job revising their paper, but some new unrealistic statements have been introduced that need to be worked on before acceptance. The change in title is highly appreciated, it better reflects the content of the paper.*

*The new Fig. 2b really looks appropriate and gives me the adequate impression of the conditions your paper is focusing on.*

*Double-checking: in the abstract the bias is -46 gC m–2, and the optimzed flux is -82 gC m–2. This means that the conventional method had -82 + (-46) = -128 gC m-2 (OK - number given on p.8, line 14).*

*Introduction now starts smoothly into the topic.*

**Reply**: We thank Prof. Eugster for his thorough review of our revised manuscript. We agree with all his comments and implemented his suggestions into the new manuscript as described below. We also include a marked-up version of the manuscript showing the changes with respect to the previous revision.

*MAJOR POINTS*

**Comment**: *(1)*
*"One conceivable mechanism is that a greater roughness length could break down larger turbulence into smaller turbulence, thus shifting some of the co-spectrum toward higher frequencies." - I do not agree!*

*14/8-10: "A greater roughness could break down larger turbulence into smaller turbulence, thus shifting some of the flux co-spectrum toward higher 10 frequencies": this is unrealistic. Turbulence is generated at the low frequency end and then is transported from larger eddies to smaller eddies and on, until turbulence dissipates in the very high frequency. In my view it is physically impossible that "greater roughness could break down larger turbulence into smaller turbulence". There is a continuity in the TKE transport from large eddies to the inertial subrange, no breakdown. Please rephrase!*

**Reply**: OK, we removed this sentence to avoid confusions and keep the focus of the discussion. The sentence thereafter was adapted such that this part now

reads: "Snowmelt also entails a change in the typical surface roughness length, which is slightly smaller for snow than open tundra vegetation. However, such shifts would be no problem for the functionality of the used flux calculation schemes and cannot readily explain bi-directional fluxes even if the surface roughness is spatially heterogeneous."

**Comment**: (2)
*2/14: "the averaging operation and coordinate rotation commonly applied in EC flux calculations 15 can lead to co-spectral distortions and a loss of flux" - my interpretation of the Finnigan et al. (2003) paper is that they specifically address this topic in tall (forest) vegetation whereas they explicitly state "This failure to close the energy balance is less common close to the surface over short roughness but is still sometimes seen, especially in complex topography." I think it cannot implicitly assumed that what they write about tall vegetation can be transferred to short-statured vegetation, thus I would add to the statement on lines 14/15 that this applies to tall vegetation, but not automatically also to tundra vegetation (I know that this was added due to Lars Kutzbach's review; but it would be good to clarify such important differences between short-statured and tall vegetation). Or has anyone shown that the Finnigan et al. (2003) findings can unanimously be applied to tundra vegetation? In my view it has to do with the fact that over forests we measure in the roughness sublayer, whereas in tundra we measure well above the roughness sublayer.*

**Reply**: We thankfully accept this suggestion to further refine this part of the introduction. The revised sentence now reads: "Finnigan et al. (2003) showed that the averaging operation and coordinate rotation commonly applied in EC flux calculations can lead to co-spectral distortions and a loss of flux in measurements over tall canopies (forests), even though it is not evident that the same holds for the short-statured vegetation of the tundra where one measures well above the roughness sublayer. Other difficulties relate to the employed measurement instruments, such as when open-path gas analyzers are used, which can introduce a bias due to surface heating of the instrument itself (Burba et al., 2008)."

*DETAILS*

**Comment**: 4/14, 5/24: avoid nested parentheses with citation
**Reply**: OK, we rephrased these sentences to remove the outer set of parentheses.

**Comment**: 5/1: the 10^-1.5 Hz number looks odd. Write "around 0.032 Hz", please. This is actually the "energy-containing range" in micrometerological terminology. Maybe use this terminology as well (if it is what your tool does).
**Reply**: OK, we changed this. The sentence now reads: "Ogive optimization furthermore only accepts periods with a negative momentum flux (i.e. directed toward the ground surface) in the mid-frequency range (tested at around 0.032 Hz), which is the energy-containing range."

**Comment**: *7/7: "which does, however, not mean that 25 s are in general enough to determine a 30 min flux" - why "in general"? Under which cases are 25 s sufficient? Would be interesting to know! If you however agree to remove "in general" then the issue could be solved in this way as well. You basically want to say that an average of multiple 25-second segments would be enough to estimate the flux, but not a single 25-second segment. I am still not 100% confident about this, to be honest.*

**Reply**: We agree with the reviewer and removed the "in general", also because this sentence is not so much meant to give a mechanistic explanation, but more to note an observation for this specific flux example.

[revised manuscript text omitted]